# A modality-independent proto-organization of human multisensory areas

Francesca Setti [1], Giacomo Handjaras [1], Davide Bottari [1], Andrea Leo [2], Matteo Diano[3], Valentina Bruno [4], Carla Tinti [3], Luca Cecchetti [1], Francesca Garbarini [4], Pietro Pietrini [1] & Emiliano Ricciardi [1] ✉

The processing of multisensory information is based upon the capacity of brain regions, such as the superior temporal cortex, to combine information across modalities. However, it is still unclear whether the representation of coherent auditory and visual events requires any prior audiovisual experience to develop and function. Here we measured brain synchronization during the presentation of an audiovisual, audio-only or video-only version of the same narrative in distinct groups of sensory-deprived (congenitally blind and deaf) and typically developed individuals. Intersubject correlation analysis revealed that the superior temporal cortex was synchronized across auditory and visual conditions, even in sensory-deprived individuals who lack any audiovisual experience. This synchronization was primarily mediated by low-level perceptual features, and relied on a similar modality-independent topographical organization of slow temporal dynamics. The human superior temporal cortex is naturally endowed with a functional scaffolding to yield a common representation across multisensory events.

The ability to combine signals across different sensory modalities is essential for an efficient interaction with the external world. To this end, the brain must detect the information conveyed by different sensory inputs and couple coherent events in space and time (that is, solve the correspondence problem[1]). Specifically, when processing audiovisual information, signals from sight and hearing converge onto multiple brain structures and, among them, the superior temporal cortex is acknowledged as being a pivotal hub[2,3]. Evidence exists that basic multisensory processing is already present in newborns[4], while audiovisual experience appears to be critical for the development of more complex multisensory computations lifelong[5,6]. Nonetheless, the extent to which audiovisual experience is a mandatory prerequisite for the superior temporal cortex to develop and become able to detect shared features between the two sensory streams is still undefined. Adult individuals who specifically lack visual or auditory input since birth represent an optimal model to test whether brain computations require a complete audiovisual experience to develop[7,8].

In this Article, we determined the synchronization of brain responses in two groups of sensory-deprived (SD, that is, congenitally blind and deaf) adults and in three samples of typically developed (TD) individuals exposed to the audiovisual, audio-only or visual-only version of the same long-lasting narrative. This approach, called intersubject correlation (ISC) analysis, postulates that brain regions synchronize across individuals when processing the same stimulus features[9]. Therefore, any evidence of synchronization within the superior temporal cortex across conditions and experimental groups would be indicative that this region yields shared representations of visual and auditory features despite so different postnatal sensory experiences. Furthermore, we provided a thorough description of the events occurring across the visual and auditory streams by developing a model-mediated version of ISC. This approach determined whether brain synchronization resulted from the processing of coherent low-level visual (for example, motion energy) and acoustic (for example, spectral properties) features, or it was instead driven

[1]MoMiLab, IMT School for Advanced Studies Lucca, Lucca, Italy. [2]Department of Translational Research and Advanced Technologies in Medicine and Surgery, University of Pisa, Pisa, Italy. [3]Department of Psychology, University of Turin, Turin, Italy. [4]Manibus Lab, Department of Psychology, University of Turin, Turin, Italy. ✉e-mail: emiliano.ricciardi@imtlucca.it

by high-level semantic (for example, language and story synopsis) characteristics. Finally, additional analyses characterized the temporal dynamics of the synchronization across individuals and depict the chronotopic organization of multisensory regions.

As expected, the activity of the superior temporal cortex was synchronized across auditory and visual inputs in TD participants. Crucially, the synchronization was also present across SD individuals, despite the congenital lack of any auditory or visual input since birth and the distinct postnatal experiences. Furthermore, the synchronization was mediated by low-level perceptual features in both TD and SD groups and relied on a similar modality-independent topographical organization of temporal dynamics consisting of adjacent cortical patches tuned to specific receptive windows. Altogether, these observations favour the hypothesis that the human superior temporal cortex is naturally endowed with a functional scaffolding to yield a common neural representation across coherent auditory and visual inputs.

## Results

ISC analysis[9] was used to measure the similarity in the brain responses elicited by the processing of either the audiovisual, the auditory or the visual streams of the same naturalistic narrative (that is, the live-action movie '101 Dalmatians') presented to both TD and SD participants by means of three functional magnetic resonance imaging (fMRI) experiments. In addition to the overall measure of synchronization provided by the ISC approach, we built a hierarchical set of models describing the low-level and the high-level features of the movie auditory and visual streams to test which stimulus properties mediate the interaction across senses. Finally, the temporal properties (that is, the temporal receptive window[10]) of the dynamical processes responsible for the synchronization across senses were studied and compared in the three experiments.

In a first experiment, the neural correlates of the audiovisual (AV) stimulus were studied in a sample of TD participants to establish how the brain processes multisensory information. In a second experiment, two unimodal versions of the same movie (that is, visual-only (V) and auditory-only (A)) were created by excluding one or the other sensory channels. Then, to investigate to what extent the neural representation of the same narrative is shared across sensory modalities, the similarity between visually and auditory-evoked brain responses (A versus V) was evaluated by performing ISC analysis across the two samples of TD participants who were exposed to either A or V conditions. Finally, in a third experiment, we studied the brain synchronization between blind and deaf participants (A versus V), listening to the audio-only (A) and watching the visual-only (V) movie, respectively.

### Brain synchronization between vision and hearing in TD

In the first experiment (Fig. 1a), a whole-brain ISC analysis was computed on TD participants exposed to the audiovisual version of the narrative (that is, multisensory condition, within-condition ISC, $N = 10$). The statistical significance of synchronization maps was based on non-parametric permutation tests and family-wise-error correction (FWEc) was applied ($P < 0.05$, one-tailed test). As shown in Fig. 2a, results highlighted a set of regions involved in the processing of multisensory information, encompassing a large extent of the cortex (~40% of the cortical volume). Significant synchronized regions included primary sensory regions, such as early auditory and visual areas, as well as high-order cortical areas, such as the superior temporal gyrus (STG) and superior temporal sulcus (STS), the inferior parietal region, the precuneus, the posterior and anterior cingulate cortex (PostCing and AntCing, respectively), the inferior frontal gyrus, and the dorsolateral and dorsomedial portions of the prefrontal cortex. The ISC peak was found in the central portion of the left STG (peak $r = 0.452$, 95th percentile range 0.206 to 0.655; Montreal Neurological Institute brain atlas coordinates (x,y,z) ($MNI_{xyz}$) −65,−14,1).

The second experiment measured the interplay between vision and audition in two groups of TD individuals ($N = 10$ A-only, $N = 10$ V-only; Fig. 1a). Synchronization (A versus V, across-modalities ISC, Fig. 2b) was present in the ~14% of the cortical volume with no involvement of primary auditory and visual areas. Significant regions comprised the superior portion of the temporal lobe, inferior parietal, precuneus, cingulate and prefrontal cortical areas. As in the case of the AV modality, the synchronization across A-only and V-only conditions peaked in the left central portion of STS (peak $r = 0.214$, 95th percentile range 0.054 to 0.485; $MNI_{xyz}$ −65,−32,1).

The brain areas identified from the above experiments in TD participants were targeted in a conjunction analysis ($P < 0.05$, one-tailed, FWEc; Fig. 2c) to highlight regions synchronized during the multisensory experience, which also shared a common neural representation across the A-only and V-only conditions. To provide a finer anatomo-functional characterization of brain regions included in the conjunction map, we adopted the Human Connectome Project parcellation atlas[11] (for a detailed description of cortical labels, please refer to Supplementary Table 1). The map (Figs. 2c and 3b) identified a set of six cortical regions, which were commonly recruited across the A-only, V-only and multisensory experimental conditions in TD participants. Specifically, the highest degree of spatial overlap was found in a bilateral temporo-parietal cluster, which comprised the superior temporal cortex (A4, A5, STSd, STSv, STGa, TGd, STV and PSL), portions of the ascending branch of the inferior temporal gyrus (PHT), the temporo-parieto-occipital junction (TPOJ) and the inferior parietal cortex (PG, PF and IP). Two additional bilateral clusters were identified: the first located in the posterior parietal cortex, comprising the PostCing, the parieto-occipital sulcus (POS) and the superior parietal (corresponding to Brodmann area (BA) 7), and the second in the medial prefrontal cortex, enclosing the bilateral AntCing and the dorsomedial portions of the superior frontal gyrus (BA 9). Lastly, two lateralized clusters were found in the left inferior frontal gyrus (BA 44 and 45) and in the right prefrontal cortex (BA 46 and 47). Of note, the conjunction map did not reveal any early sensory areas indicating that the activity of those regions was not synchronized in response to the audio-only and visual-only conditions (A versus V; Fig. 2b).

Altogether, results of experiments 1 and 2 showed that well-known multisensory areas[2] synchronize to audiovisual correspondences over time, even when sensory features are provided unimodally.

### Brain synchronization between vision and hearing in SD

In the third experiment, ISC analysis tested the similarity of brain responses across V-only and A-only conditions in congenitally deaf and blind participants, respectively. Specifically, across-modality ISC (that is, A versus V; Fig. 1a) was performed in nine blind and nine deaf individuals. In this experiment, ISC was computed in the regions of interest identified by the conjunction map obtained from the first two experiments with TD participants.

As shown in Fig. 3a, congenital lack of either auditory or visual experience did not prevent synchronization of brain responses across modalities. Indeed, significant synchronization was found in the bilateral temporal cortex (A4, A5, STS, STV and STGa), TPOJ, PostCing and POS, which represented ~47% of the conjunction map identified in TD participants and 5% of the overall cortical volume ($P < 0.05$, one-tailed, FWEc). Moreover, the ISC map highlighted a minimal involvement of the bilateral inferior parietal (PG and PF), the right dorsolateral prefrontal (BA 46) (~1% of the conjunction mask), and the left inferior temporal cortex (PHT and PH; ~1%). Notably, SD individuals showed the ISC peak within the central portion of left STS similarly to TD participants (peak $r = 0.131$, 95th percentile range −0.027 to 0.372; $MNI_{xyz}$ −6,−32,1). On the contrary, SD groups did not show any significant synchronization in bilateral medial prefrontal (AntCing and BA 9), left inferior frontal (BA 44 and 45) and prefrontal cortex (areas 46 and 47). To further investigate the consistency of synchronization at single subject level, the

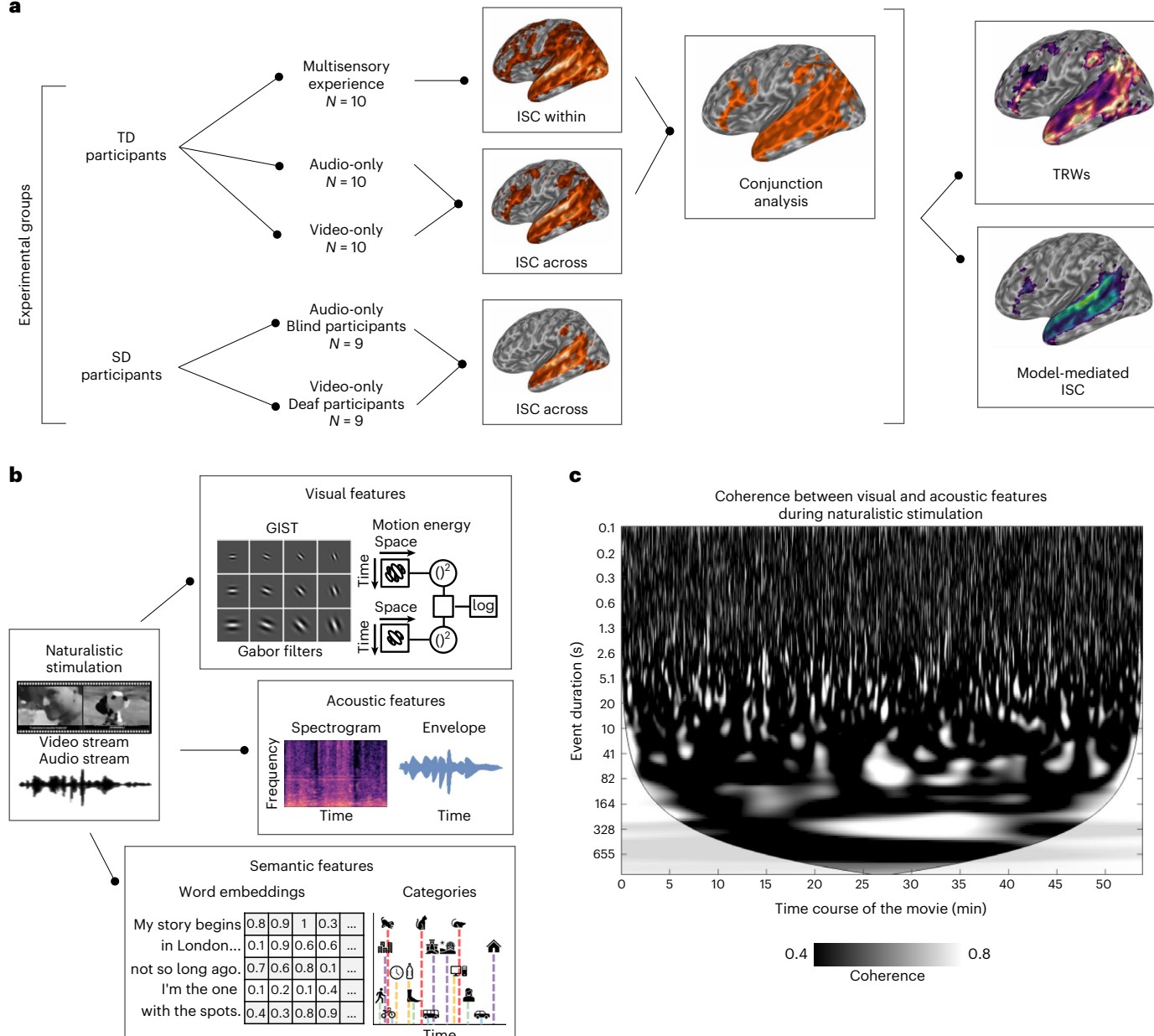

**Fig. 1 | Experimental conditions, computational modelling and analytical pipeline. a**, In the first experiment, the neural correlates of a full audiovisual (AV) stimulus were studied in a sample of TD participants to examine how the brain processes multisensory information. In a second experiment, two unimodal versions of the same movie (that is, visual only (V) and auditory only (A)) were presented and the similarity across visually and auditory-evoked brain responses (A versus V) was assessed in two samples of TD participants. In a third experiment, we tested the role of audiovisual experience for the emergence of these shared neural representations by measuring the similarity of brain responses elicited across congenitally SD individuals (that is, blind and deaf participants). **b**, A brief description of the features extracted through computational modelling from the movie. Movie-related features fall into two categories: (1) low-level acoustic (for

example, spectral and sound envelope properties to account for frequency- and amplitude-based modulations) and visual features (for example, set of static Gabor-like filters and motion energy information based on their spatiotemporal integration); and (2) high-level semantic descriptors (for example, manual annotation of both visual and sound-based natural and artificial categories and word embedding features; for further details, see Supplementary Information). In **c**, we show the results of a continuous wavelet transform analysis applied to the movie acoustic and visual signals to evaluate the existence of collinearities across the low-level features of the two sensory streams. Results show the presence of cross-modal correspondences, with hundreds of highly coherent events (white marks) distributed along the time course of the movie (x axis), lasting from a few tenths of a second to several minutes (y axis).

raw ISC was computed across all participants in a region of interest defined in the left STG (Fig. 3a). The ISC matrix confirmed high synchronization between subject pairings of the audio-only conditions, in line with the role of the temporal cortex in auditory computations. Additionally, spread synchronizations emerged between the individuals exposed to the visual-only condition and all the other participants,

supporting the hypothesis of a modality-independent processing of information in this cortical patch.

The comparison between TD and SD participants in the A versus V condition (Fig. 3b) showed a diffuse decrease of ISC across all the explored regions (Wilcoxon rank sum test, $P < 0.05$ two-tailed, Bonferroni corrected), with the notable exception of the posterior parietal

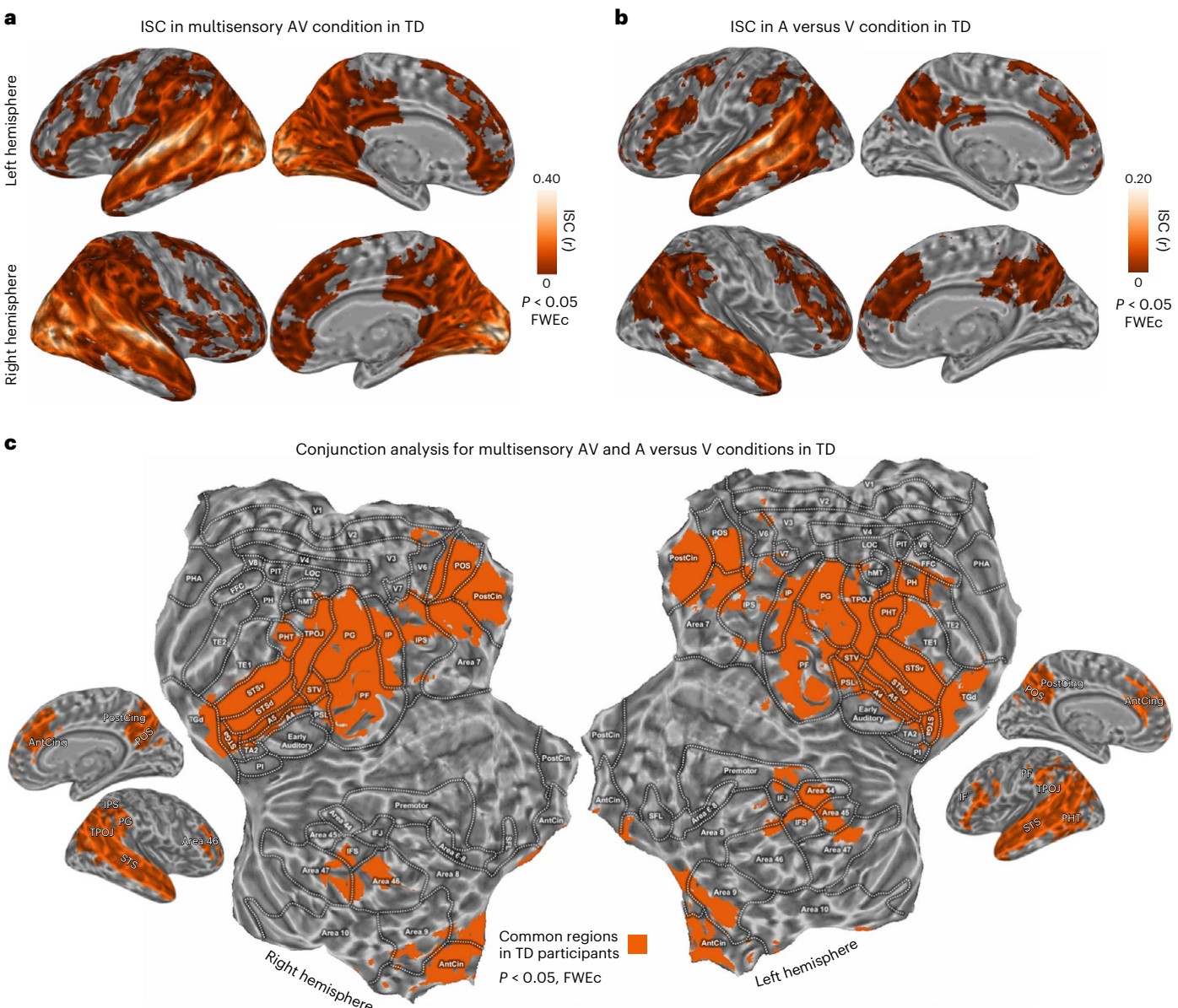

**Fig. 2 | ISC results in TD participants. a,b,** ISC in TD participants in the AV (**a**) and A versus V (**b**) conditions respectively (*P* < 0.05, one-tailed, FWEc, minimum cluster size of 20 adjacent voxels). **c,** Conjunction analysis of the two aforementioned experimental conditions.

cortex ($P_{bonf}$ = 0.085). Moreover, in SD participants, the left inferior frontal gyrus and the bilateral medial prefrontal cortex were not statistically synchronized, with averaged ISCs falling within the null distribution.

Altogether, these results indicated that congruent auditory and visual streams elicited a functional synchronization in the superior temporal areas and in the postero-medial parietal cortex even in the case of congenital auditory or visual deprivation and, thus, in absence of prior audiovisual experience.

### The role of perceptual and semantic stimulus features

Brain activity of congenitally deaf and blind people was synchronized when exposed to the same narrative. Nevertheless, whether the synchronization could be ascribed either to the processing of perceptual (that is, low-level) features, or to semantic (that is, high-level) representations shared across the different conditions, remained to be determined. We took advantage of computational modelling (Fig. 1b) to extract fine-grained, low-level features from both the auditory (for example, spectral and sound envelope properties to account for frequency- and

amplitude-based modulations) and visual streams (for example, set of static Gabor-like filters and motion energy information based on their spatiotemporal integration). Moreover, a set of high-level features was collected by means of manual annotation and automated machine learning techniques (for example, word embedding) to represent the linguistic and semantic properties of the narrative.

To address the role of low-level visual–acoustic and high-level semantic features, we measured the impact of each model in modulating the magnitude of the ISC between the unimodal conditions (Fig. 4; *P* < 0.05, one-tailed, FWEc). Specifically, each model was regressed out from the individual brain activity before computing the ISC. This procedure resulted in a reduction of the ISC value that reflects, for each brain region, the relevance of distinct stimulus descriptors that were regressed out. Therefore, the role of each model as a *mediator* was evaluated on the synchronization of brain responses across participants and the relative drop in the ISC magnitude was computed for each voxel[12]. In principle, if a model explains entirely the activity of a specific brain region, the ISC in that area will drop substantially,

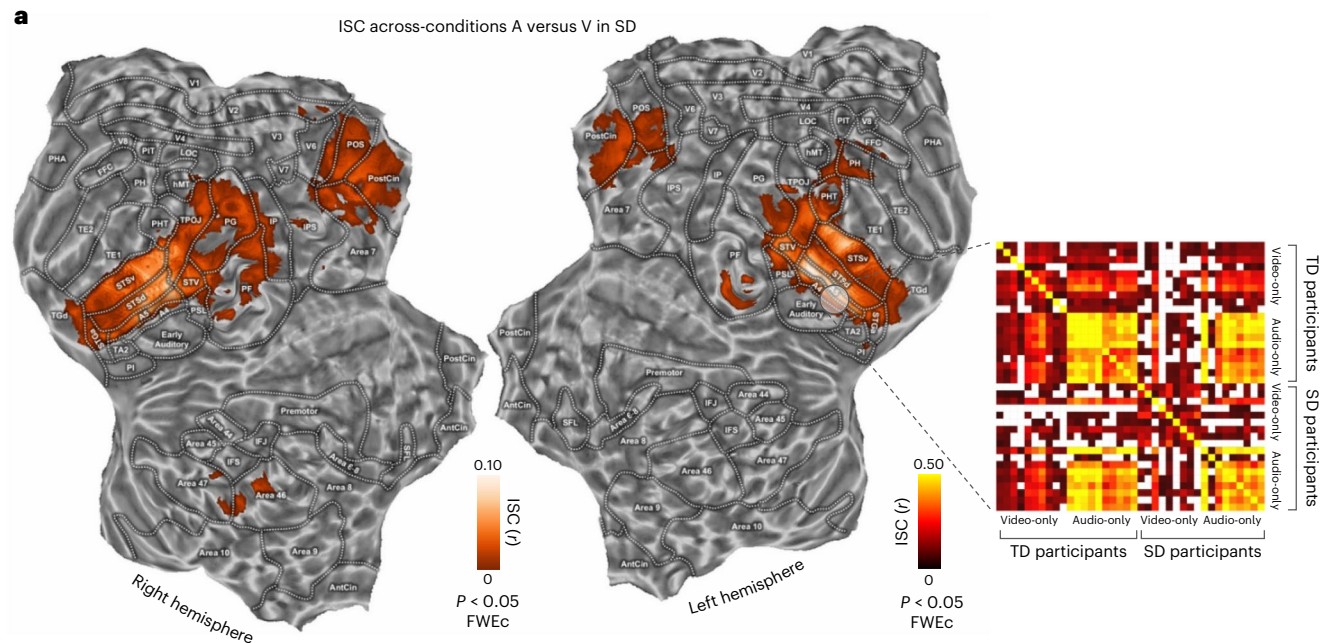

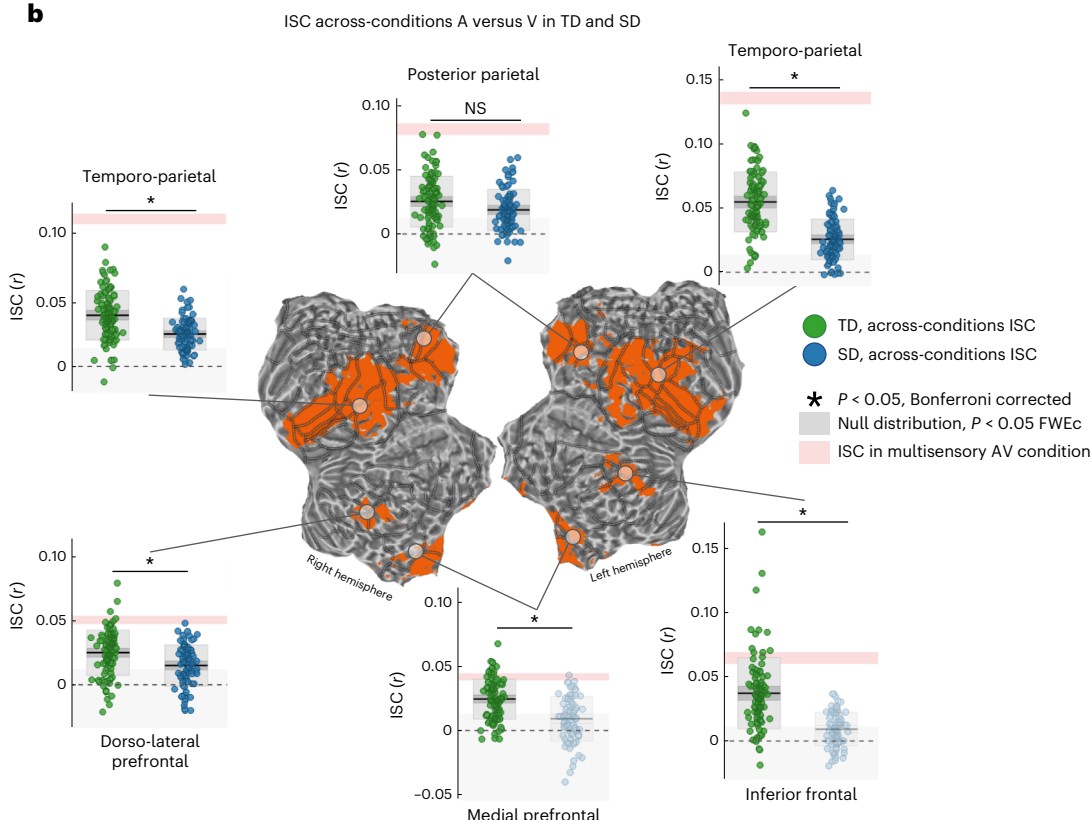

**Fig. 3 | ISC results in SD participants. a**, ISC results from the across-modality A versus V in SD participants ($P < 0.05$, one-tailed, FWEc, minimum cluster size of 20 adjacent voxels). In the matrix on the right, the raw ISC across TD and SD individuals was reported for the A-only and V-only conditions. ISC (Pearson's $r$ coefficient) was extracted from a region of interest (6 mm radius) centred within the left STG, around the synchronized peak of the first experiment. White cells indicated subject pairings below the significant threshold (uncorrected $P < 0.05$, one-tailed). **b**, ISC for the A versus V condition was compared between TD and SD participants within the six brain regions obtained from the conjunction analysis (Wilcoxon rank sum test, two-tailed, $P < 0.05$ Bonferroni corrected for the number of regions). All regions except the bilateral posterior parietal cortex retained a significantly greater ISC in TD than SD individuals (left temporo-parietal: $W = 11,926$, $P_{Bonf} < 0.001$, $N_{TD} = 100$,

$N_{SD} = 81$, $r_{TD-SD} = 0.029$, standard error (SE) 0.003; bilateral posterior parietal: $W = 9,961$, $P_{Bonf} = 0.085$, $N_{TD} = 100$, $N_{SD} = 81$, $r_{TD-SD} = 0.007$, SE 0.003; right temporo-parietal: $W = 11,077$, $P_{Bonf} < 0.001$, $N_{TD} = 100$, $N_{SD} = 81$, $r_{TD-SD} = 0.015$, SE 0.002; right dorso-lateral prefrontal: $W = 10,452$, $P_{Bonf} = 0.001$, $N_{TD} = 100$, $N_{SD} = 81$, $r_{TD-SD} = 0.010$, SE 0.002; bilateral medial prefrontal: $W = 11015$, $P_{Bonf} < 0.001$, $N_{TD} = 100$, $N_{SD} = 81$, $r_{TD-SD} = 0.015$, SE 0.002; left inferior frontal: $W = 11,913$, $P_{Bonf} < 0.001$, $N_{TD} = 100$, $N_{SD} = 81$, $r_{TD-SD} = 0.028$, SE 0.003). Average ISC (with SE) in the AV condition is shown, by a shaded area in rose, as a ceiling effect due to multisensory integration. Transparency is applied to indicate that the group ISC was not significant (NS, $P > 0.05$) compared with a null distribution. In each box, the dark line represents the sample mean and the dark-grey shaded box the 95% confidence interval of the SE of the mean, while the light-grey shaded box indicates the standard deviation.

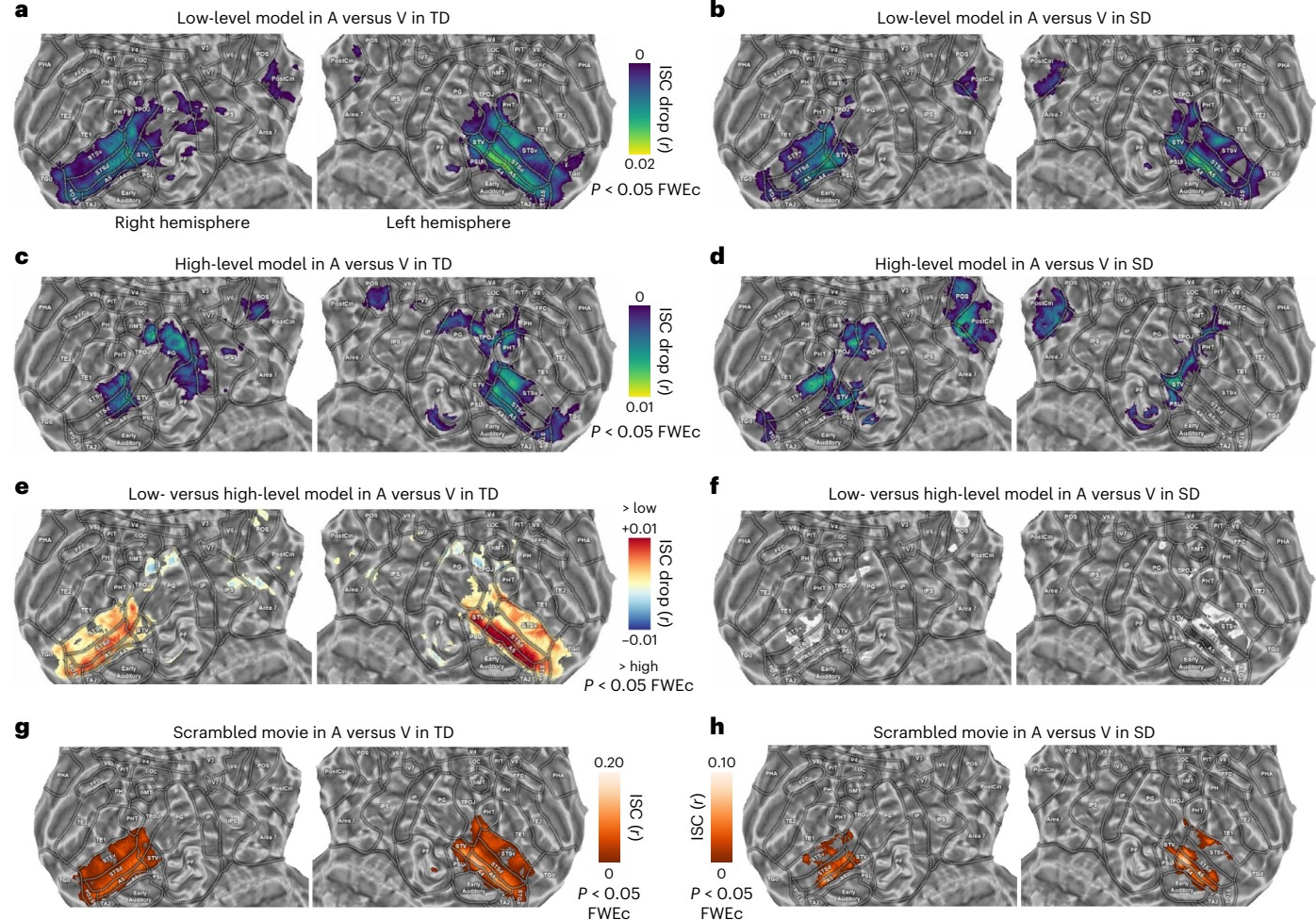

**Fig. 4 | Impact of perceptual and semantic features on ISC. a,b,** Model-mediated ISC across TD and SD participants in the A versus V condition for the low-level model, based on the movie acoustic and visual properties ($P < 0.05$, one-tailed, FWEc, minimum cluster size of 20 adjacent voxels). **c,d,** Model-mediated ISC across TD and SD individuals in the A versus V condition for the high-level model, based on semantic features (that is, categorical information and GPT-3 features, $P < 0.05$, one-tailed FWEc, minimum cluster size of 20

adjacent voxels). **e,f,** Results of a Wilcoxon signed rank test comparing the low- and high-level models in TD and SD participants separately ($P < 0.05$, two-tailed, FWEc). **g,h,** ISC for the groups of TD and SD individuals in the A versus V condition during the processing of the scrambled movie ($P < 0.05$, one-tailed, FWEc, minimum cluster size of 20 adjacent voxels). Note that only the temporo-parietal cortex is mapped, since we do not find any significant results in frontal areas for the SD group in any of the explored conditions.

with values that will approach zero. Conversely, if a model does not contribute to the synchronization of brain responses across visual and auditory movie processing, the drop in the ISC magnitude will be negligible. We named this approach model-mediated ISC.

Concerning the low-level models, we regressed out visual features from brain activity during the A-only stimulation and acoustic features from the activity during the V-only processing to test whether a drop of ISC magnitude could be ascribed to audiovisual correspondences. Thus, this procedure identifies the impact of the unique portion of model variance shared across the two modalities. In both TD and SD groups the drop of ISC was significant in the posterior parietal and STS/STG regions (A4, A5 and STSd), and maximum in the central portion of the left STS (Fig. 4a,b; drop of ISC at peak: TD A versus V: $r = 0.020$, 95th percentile range 0.006 to 0.039, MNI$_{xyz}$ −62,−26,1; deprived A versus V: $r = 0.018$, 95th percentile range −0.014 to 0.061, MNI$_{xyz}$ −65,−26,1). Consequently, these cortical areas retain a low-level representation of audiovisual features that inherently co-occur in a naturalistic stimulation.

Concerning the role of language and semantics, we combined the features generated by the representation of sentences and those extracted by manual annotation of categories. Therefore, these

high-level features, which are naturally multimodal, were removed from the brain activity of all participants. Results of the model-mediated ISC revealed that semantic features had a significant impact in synchronization across modalities in the posteromedial parietal cortex in both TD and SD participants. As regards the temporal cortex, in SD participants only the STV and the posterior portions of STS were affected by semantic features, whereas in TD participants model influence was spread across the whole STS, and particularly in the left hemisphere (Fig. 4c,d; drop of ISC at peak: TD A versus V in left MTG: $r = 0.011$, 95th percentile range −0.007 to 0.026, MNI$_{xyz}$ −53,−71,1; deprived A versus V in the right TPOJ: $r = 0.011$, 95th percentile range −0.009 to 0.032, MNI$_{xyz}$ 49,−74,19).

Finally, we tested whether model-mediated ISC differed between the low-level model and the high-level one (Fig. 4e,f). Results in both TD and SD groups indicated that the mediation of low-level features had a higher magnitude ($P < 0.05$, two-tailed, FWEc) in posterior and middle STS/STG, whereas, in their anterior portions, this effect was limited to TD individuals only. On the other hand, in both TD and SD groups, the high-level features exhibited a significantly higher effect ($P < 0.05$, two-tailed, FWEc) in a small patch of cortex, centred around bilateral TPOJ, PG and POS regions.

These results demonstrate that, in both TD and SD, STG/STS synchronization was primarily driven by lower-level properties.

## The impact of the movie plot

Model-mediated analyses clarified the contribution of low- and high-level features to brain synchronization. However, this approach did not test the extent to which the temporal sequence of connected events in the narrative (that is, story synopsis) determines brain synchronization. To account for this possible mechanism, a control condition was based on a scrambled short movie in which we manipulated the chronological order of previously unseen cuts (lasting from 1 to 15 s, median: 3.5 s, maintaining the same distribution of cut lengths of the original movie). Because of this manipulation, even though the storyline of the control condition was nonsensical, a set of stimulus features were preserved: (1) the cinematography, with the same coarse- and fine-grained visual features, (2) the sound mixing, (3) a semantic representation based on single words up to very short sentences and (4) the editing pacing. Importantly, we left untouched the congruency between audio and visual streams, which remained synchronized with each other.

Interestingly, the scrambling of the movie plot differentially affected the synchronization across brain areas ($P < 0.05$, one-tailed, FWEc; Fig. 4g,h). Indeed, a meaningless narrative still triggered shared responses in the central and posterior parts of superior temporal cortex (A versus V in TD and SD), particularly in A4, A5 and STSd, and the peak was located in the central part of the left STS, with similar intensities as compared with the original movie (ISC at peak: TD A versus V: $r = 0.162$, 95th percentile range $-0.004$ to $0.383$, $MNI_{xyz}$ $-64, -14, -4$; SD A versus V: $r = 0.105$, 95th percentile range $-0.050$ to $0.397$, $MNI_{xyz}$ $-64, -32, 2$). Remarkably, the disruption of the narrative significantly affected the posteromedial parietal regions, whose synchronicity did not reach the significance threshold in both TD and SD individuals.

Therefore, this further evidence confirms that synchronous correlations in specific portions of the temporal cortex (A4, A5 and STSd) were primarily driven by low-level perceptual properties and not by high-level semantic computations required for the processing and understanding of the narrative.

## Temporal dynamics across vision and hearing

Additional analyses were conducted to characterize the temporal properties of the synchronization across individuals. First, we evaluated the correspondences between the auditory and visual streams in our naturalistic stimulation. To compare visual (that is, pixel intensities) and acoustic (that is, sound wave energy) information, a set of descriptors (that is, static Gabor-like filters and spectral features) were extracted at the highest available sampling frequency (25 Hz, the frame rate of the original movie). Afterward, the coherence in time of the two streams was measured by means of a continuous wavelet transform to detect both the duration and the onset time of specific events in the movie, shared across auditory and visual streams. Although a set of relatively coarse computational features were used, the results reported in Fig. 1c demonstrated the existence of a multifaceted series of highly coherent events, lasting from tenths of a second to several minutes. Considering both the high variability in temporal dynamics of the correspondences across modalities in the movie, and the limited temporal resolution of fMRI, we expected that brain regions would have expressed different temporal tunings from tens to hundreds of seconds to process the incoming sensory input. To address this point, we estimated the ISC in our regions of interest by means of a temporal filter of increasing window widths, from one timepoint (that is, 2 s, which led to the same results of a classical ISC pipeline), up to 4 min. Being conceptually analogous to other techniques[10], this methodological approach was defined as temporal receptive window (TRW) analysis. We estimated both the length of the temporal window that retained the highest ISC (Fig. 5, left) and the temporal profile of ISC for all the explored window widths (Fig. 5, right). A high ISC over a short window (that is, few seconds) would suggest that the correlation was modulated by rapidly changing events, whereas high ISC values over longer segments (that is, tens of seconds) indicated that the correlation mostly relied on accumulating information.

TRW results further highlighted the role of the superior temporal and posteromedial parietal cortex ($P < 0.05$, one-tailed, FWEc) in detecting commonalities across multiple sensory streams. Specifically, coherent TRW maps across the three experimental conditions were found in the superior temporal cortex: A4, A5 and STS exhibited a patchy organization[13], in which adjacent subregions demonstrated distinct peak responses of temporal preference (Fig. 5a–c, left). In detail, ISC in A4, A5 and STSd showed a selective tuning for the fastest synchronization timescale, peaking at 10–20 s, whereas the middle and anterior portions of the sulcus, particularly in their ventral part, displayed a preference for timescales longer than 1 min. Moreover, synchronization in the PostCing cortex and the POS was characterized mostly by slow modulations with an average preferred response to events occurring about every minute. To test the spatial similarity between TRW maps, we measured their correlations in the voxels that resulted to be significant in the A versus V condition in SD participants. Results demonstrated that temporal tunings were coherently represented across all the three experimental conditions, showing high correlations between maps (between A versus V in TD and SD: $\rho = 0.334$, 95th percentile range 0.290 to 0.380; between A versus V in TD and AV in TD: $\rho = 0.431$, 95th percentile range 0.392 to 0.473; between A versus V in SD and AV in TD: $\rho = 0.402$, 95th percentile range 0.361 to 0.443; all correlations with $P < 0.005$, two-tailed).

To summarize, the overall TRW profile was consistent across the three experimental procedures, indicating that synchronized regions retained a similar chronotopic organization.

## Discussion

The present study tested whether brain regions involved in audiovisual processing, including the superior temporal and neighbouring areas, retain the ability to represent sensory correspondences across modalities, despite the complete lack of prior auditory or visual experience. To this purpose, we characterized the coherence of neural responses evoked during a naturalistic stimulation across different modalities and across independent samples of TD, congenitally blind and congenitally deaf individuals by performing three distinct fMRI experiments. Results indicate that a functional architecture of the superior temporal cortex, based on the extraction of common basic features from auditory and visual signals, emerges despite the lack of audiovisual inputs since birth and thus irrespectively of postnatal audiovisual sensory experiences. This observation favours the hypothesis that the human superior temporal cortex is endowed with a functional scaffolding to process low-level perceptual features that define sensory correspondences across audition and vision.

The ISC analyses were exploited to evaluate shared representations across visually and acoustically evoked responses in both TD and congenitally SD individuals.

Overall, the three experiments revealed a set of regions—STS/STG and neighbouring areas—synchronized over time by correspondences between visual and auditory inputs. The congenital absence of visual or auditory experience does not affect the ability of the superior temporal cortex, a brain region devoted to integrated audiovisual processing, to represent unimodal auditory and visual streams pertaining to the same perceptual events. Specifically, high ISC values in temporal and parietal regions were observed for the audiovisual stimulation as a whole. Moreover, when measuring the correlation across the two unimodal conditions, the synchronization was maintained not only in TD participants but also among the subjects from the two SD groups. In summary, a shared synchronization demonstrates that the ability of these areas to process signals originating from the same natural

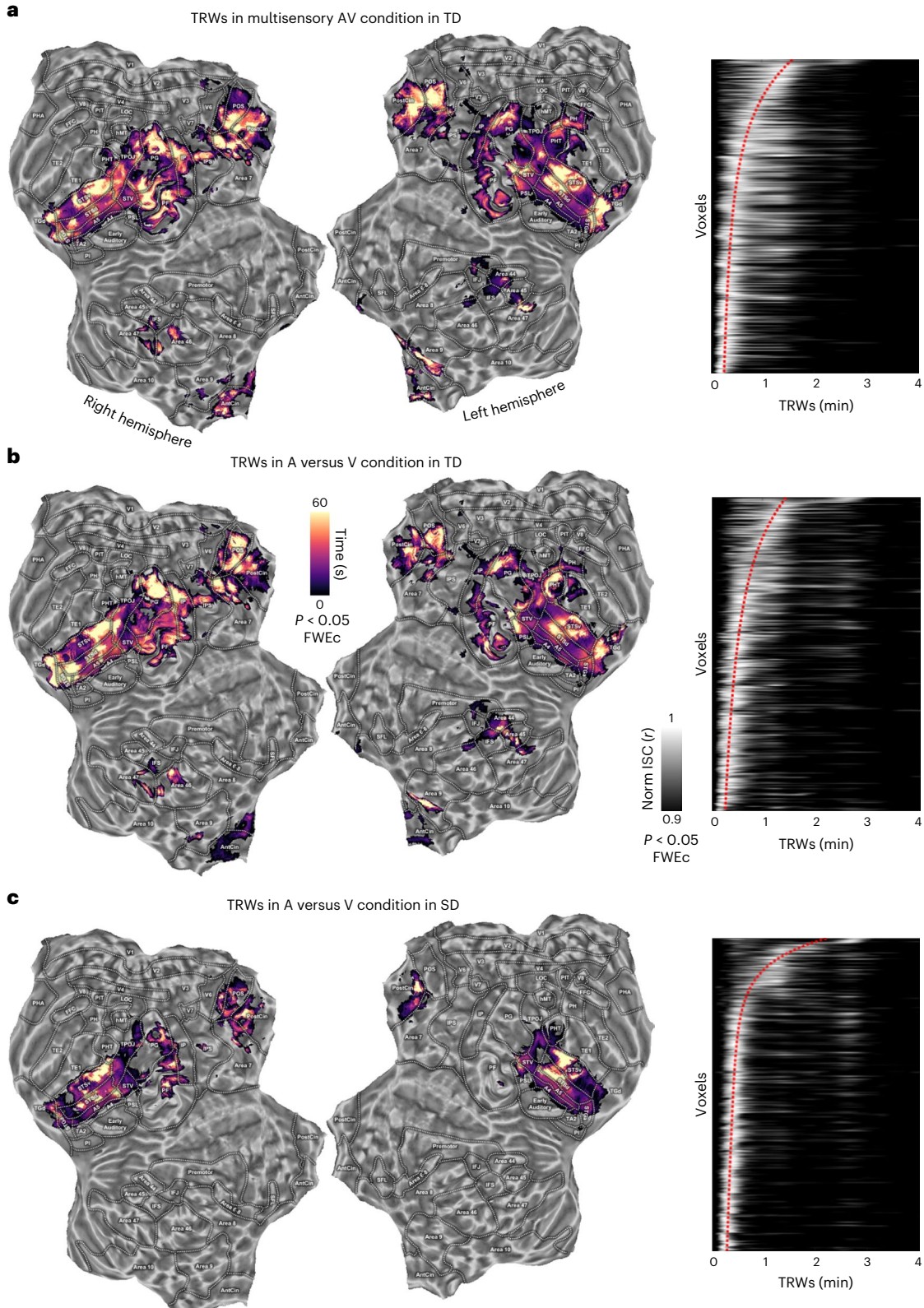

**Fig. 5 | TRWs. a–c**, TRW results in the three experimental conditions (*a*, multisensory AV condition in TD; *b*, across-modality in TD; *c*, across-modality in SD) using windows from 2 s to 240 s. On the left, flat brain maps show the temporal peak when ISC is maximal. Matrices on the right indicate the overall synchronization profile across voxels which survived the statistical threshold (*P* < 0.05, one-tailed, FWEc, minimum cluster size of 20 adjacent voxels) in the A versus V condition in SD participants. The voxels were sorted according to the peak of the TRWs in the SD condition. Voxel order was then kept constant across the other two experiments. Pixel intensity depicts the normalized ISC (scaling the maximum to one). Red dotted lines represent the interpolated position of maximal peaks across ordered voxels. Only a few voxels presented responses characterized by multiple peaks, whereas the majority demonstrated a clear preference for a specific temporal window.

events in a modality-independent manner is maintained even with the lack of any prior audiovisual experience. However, although synchronized, the ISC magnitude in the congenitally blind and deaf groups was significantly lower than the one measured in the TD group. These differences probably result from distinct postnatal experiences and atypical developmental trajectories that may affect language processing or semantic representation[14–16].

In addition to STS/STG, auditory and visual responses also were synchronized within the posteromedial parietal areas (for example, posterior cingulate and POS) in both TD and SD. On the contrary, synchronization in the left inferior frontal gyrus and in the bilateral medial prefrontal cortex was observed in TD participants only. Moreover, scrambling the chronological order of scenes hindered the synchronization in these parietal regions, providing further evidence for their involvement in discourse and narrative understanding across modalities (Supplementary Discussion). The same procedure did not impair synchronization in the superior temporal cortex, suggesting that other features are more relevant to this area.

Features modelling was adopted to test the computational properties of synchronized cortical regions. To weigh the relative contribution of distinct stimulus properties, here we developed a model-mediated ISC approach. To ease the interpretability of our results, we removed a set of coarse descriptors encompassing non-specific, visual, auditory and linguistic properties that were shared across all the computational models (Supplementary Results and Supplementary Discussion).

Our low-level perceptual models were inspired by previous studies showing that, during the processing of natural sounds, STS/STG extracts multiple acoustic features, including the amplitude modulation of the temporal envelope[17], the spectro-temporal modulations[18,19], the pitch[20] and the timbre[21]. Moreover, the same regions are also pivotal hubs of the language network[22]. In particular, the middle and posterior portions of STS encode phonetic features[23], whereas more anterior regions are involved in lexical and semantic processing and contribute to the grammatical or compositional aspects of language[24]. Regarding the processing of visual properties, there is compelling evidence that STS/STG represents biological motion[25]. Indeed, neurons in middle and posterior STS detect both the snapshots of a biological action and their spatiotemporal integration, as well as complex visual patterns of motion[25]. Taken together, the ability to code static and dynamic properties of visual stimuli, combined with the capacity to process acoustic information and to integrate the two modalities[1], allows STS/STG to encode multisensory objects[2], to solve face–voice matching[26], to represent actions[27,28] and, more in general, to respond to biologically salient events[1].

In our study, the ISC of STS/STG was significantly moderated by spectral and amplitude modulations of sounds, and by static Gabor-like filters and motion energy of visual input in both TD and SD individuals. This suggests that low-level properties of both auditory and visual inputs exert a crucial role for the computations performed by these regions. This observation is in line with previous evidence showing that the dynamic properties of visual and acoustic signals during speech processing (that is, lip contour configuration and sound envelope) are correlated and drive stimulus perception, as it occurs in the McGurk effect[29].

Regarding high-order properties, either defined by means of sentence embeddings or categories, previous evidence suggested the involvement of STS/STG in semantics[30–32]. Other studies proposed that semantic representations may instead retain a large-scale organization distributed throughout the cortex[22,33]. Here we observed that the high-level movie properties, such as contextual word embeddings and categorical semantic information, mediate the synchronization across auditory and visual stimulations in TD but not in SD participants. While perceptual features are intrinsic properties of a given stimulus, higher-order characteristics rely more on experience and learning, and this may partially account for the observed differences between TD and SD participants. Additionally, idiosyncratic processes of neural re-organization as a result of a specific form of sensory loss may hamper synchronization among SD individuals. For instance, studies show that the superior temporal cortex undergoes postnatal functional re-organization in congenitally deaf people during linguistic processing[34–36]. Similarly, congenital loss of visual input in blindness affects language processing and lateralization of language functions[15,37].

When measuring the temporal characteristics of the correspondences between visual and acoustic features in the movie, a series of highly coherent events, lasting from tenths of a second to minutes, were found. Such correspondences in the stimulus properties provided the basis for the TRW analysis that estimates the temporal tuning and the duration of receptive windows in the brain. Although hampered by the limited temporal resolution of fMRI, TRW analysis revealed temporal windows ranging from ten seconds (for example, A4, A5 and STSd) to a couple of minutes (for example, middle and anterior STSv, PostCing and POS). The temporal dynamics of stimulus processing were strikingly consistent across our experimental groups and were arranged to form topographic maps, with adjacent patches of cortex showing distinct temporal tunings. Note that here the term 'topography' is used to indicate a coarse hierarchy of TRW, extending over multiple brain regions and evoked by the processing of naturalistic sensory information. These slow-scale chronotopic maps are coherent with previous studies that manipulated the temporal structure of visual[10], auditory[38] and multisensory[39] naturalistic stimulations: early primary areas exhibited shortest TRWs, whereas high-order brain areas (for example, TPOJ, PostCing, POS, precuneus and frontal regions) elaborated and integrated information that accumulated over longer timescales. Equally, when analysing how events are hierarchically processed at multiple timescales in the brain, high-order areas (including TPOJ and the posterior portions of the medial cortex) represent long-lasting, more abstract and modality-independent events[40].

As far as the large-scale topographical arrangement of temporal features in the STS/STG is concerned, literature mainly focused on the mapping of acoustic (and language-derived acoustic) properties. Specifically, an anterior-to-posterior gradient was demonstrated when processing phrases, words and phonemes, respectively[41]. Furthermore, a gradient representing features extracted from the speech envelope, as well as from synthetic stimuli, also was found in the central portion of STG[42]. Specifically, within this central portion of STG, the more anterior part represented slower sounds with relatively high spectral modulations (for example, syllabic or prosodic timescales), whereas the more posterior one encoded faster sounds with a relatively low spectral modulation (that is, phonemic timescale). Our topographical organization supports the observation that encoded information in STS/STG represents acoustic, rather than semantic, properties[42,43]. More importantly, the existence of a coherent topographical organization was demonstrated even for visual-only stimulations, favouring the hypothesis of a modality-independent functional organization in the STS/STG.

Altogether, our TRW analysis in the temporal and parietal areas endorses the possibility of a representation of correspondences across modalities based on a hierarchical temporal tuning that is maintained despite atypical postnatal trajectories of sensory development.

Individuals born with the congenital lack of a sensory modality have been offering the opportunity to understand the extent to which prior sensory experience is a mandatory prerequisite for the brain organization to develop[7,8,44,45]. Here, two distinct models of sensory deprivation were specifically studied using the same stimulus content, as conveyed through the spared sensory channel. The functional features shared between the two SD models, and then with the TD groups across unimodal and multisensory stimuli, imply that the superior temporal cortex processes congruent signals related to the same event—in a modality-independent manner and autonomously from a prior (multi) sensory experience—and is prevalently predetermined to act as a 'correlator' of perceptual information across modalities.

Previous evidence has been showing that the morphological and functional large-scale architecture of the human brain results to be largely preserved in SD models despite the lack of a sensory input since birth and is characterized—to some extent—by modality-invariant computations across several functional domains[7,8]. The evidence of this supramodal organization has been extended here also to multisensory cortical regions. As a matter of fact, observations in newborns[46] and sensory-restored individuals[47,48] already indicated that basic integration of multiple sensory inputs does not require an early multisensory experience to develop[4,49,50] and that the functional architecture needed for multisensory processing may be already present within the first weeks of life[51].

At the same time, our results also favour the hypothesis that audiovisual experience is required for a full refinement of multisensory functions within temporal and parietal regions. Specifically, in the STS/STG both the reduced synchronization and the reduced representation of higher-level features in SD individuals suggest that sensory experience is necessary for a complete development of their functional specializations. Consistently, even if our study cannot directly evaluate this refinement, audiovisual experience appears to be pivotal for the full development of higher-level computations[5,6] (for example, speech perception or semantic congruence).

Altogether, the present findings indicate that human areas subserving audiovisual processing may be innately provided with a functional scaffolding to yield basic multisensory representation and perceive coherence across sensory events. The existence of a 'proto-organization' in the multisensory areas is aligned with the morphogenetical and functional evidence that large portions of the human neocortex may possess a predetermined architecture that is based on a sensory-grounded representation of information and forms the scaffolding for a subsequent, experience-dependent functional specialization[52]. Indeed, the innate presence of a topographic organization of the visual and sensorimotor systems provides the foundations for the progressive development and refinement of vision-[53–55], somatosensory- and motor-related functions[56]. Equally, previous experimental evidence in SD models nicely matches with this hypothesis of an architecture, characterized by topographical maps already at birth and whose refinement is then favoured by the cooperation of distinct sensory inputs[52].

Nonetheless, it is important to remark that, by definition, neither the combined multisensory stimulation nor the effects of the interaction among inputs (for example, as in the McGurk effect) could be directly assessed in SD individuals. Indeed, measuring the synchronization across two modalities is not equivalent to claiming that we are measuring multisensory processing. Consequently, no specific assessment of the functionality of this 'proto-organization' towards multisensory integration could be directly verified here. Additional limitations about the sample size characteristics, the low temporal resolution of fMRI and the computational modelling are discussed in Supplementary Discussion.

To conclude, here we studied three distinct samples of TD and two models of congenital SD individuals presented with the same narrative via multisensory and unimodal streams to investigate the neural representation of sensory correspondences. The demonstration of a preserved functional topography in the superior temporal cortex favours the hypothesis of an innate, modality-independent functional scaffolding to yield basic multisensory processing. Within the old 'nature versus nurture' debate, our study sheds new light on the extent to which audiovisual experience is a mandatory prerequisite for the detection of coherent events across senses.

## Methods

### Ethics declaration

Each volunteer was instructed about the nature of the research and gave written informed consent for the participation, in accordance with the guidelines of the institutional board of Turin University Imaging Centre for brain research. The study was approved by the Ethical Committee of the University of Turin (protocol number 195874, 29 May 2019) and conforms to the Declaration of Helsinki.

### Participants

Fifty subjects took part in the study. We enroled both TD individuals and SD subjects who lack visual or auditory experience since birth.

Three samples of TD individuals underwent a different experimental condition consisting in the presentation of one version of the same movie: either (1) the full multimodal audiovisual (AV) ($N = 10$, $35 \pm 13$ years, 8 females), (2) the auditory (A) ($N = 10$, $39 \pm 17$ years, 7 females) or (3) the visual (V) ($N = 10$, $37 \pm 15$ years, 5 females) one. SD individuals comprising blind ($N = 11$, mean age $46 \pm 14$ years, 3 females) and deaf ($N = 9$, mean age $24 \pm 4$, 5 females) participants were presented with the A and V movie conditions, respectively. Two blind subjects were removed from the fMRI analysis for excessive head movement (final sample, $N = 9$, mean age $44 \pm 14$ years, 3 females). All participants were right-handed, as resulted from the scores of the Edinburgh Handedness Inventory. Blind and deaf participants were congenitally deprived with the exception of one deaf subject that reported sensorineural hearing loss before the first year of age, and have no history of any psychiatric or neurological disorders. All deaf individuals were proficient in Italian Sign Language and did not use hearing aids at the moment of the study. TD subjects reported no hearing impairment, normal or corrected-to-normal vision and no knowledge of Italian Sign Language. Only native Italian speakers were enroled in the study. Additional information about the deaf and blind samples is provided in Supplementary Table 2.

### Stimulus

Naturalistic stimulation was provided through the presentation of the V, A and AV versions of the live-action movie '101 Dalmatians' (S. Herek, Great Oaks Entertainment & Walt Disney, 1996). To facilitate subjects' compliance, a story with a linear plot was selected to make the narrative easy to follow in the context of unimodal presentation. The movie was shortened to make it suitable for a single scanning session. For this purpose, we discarded the scenes whose exclusion does not alter the main narrative thread and we merged the remaining parts together to ensure smooth transitions among cuts while preserving the continuity of narration. The movie was edited to a final version of about 54 min that was then split into six runs (~8 min). A fade-in and fade-out period (6 s) were inserted at the beginning and the end of each run. In addition, a scrambled run (8 min) was built by randomly sampling and concatenating the discarded sequences according to the distribution of the duration of movie cuts. This procedure preserved the same low-level features of the original movie while purposely disrupting the narration.

As concerns the auditory version of the stimulus, a voice-over was superimposed over the original movie soundtrack to convey the information carried by the missing visual counterpart. The Italian audio description of the movie was adapted to our shortened version of the film. Therefore, several parts of the original script were re-written not only to better bridge the gaps we introduced via editing, but also to ensure a satisfactory verbal depiction of those aspects of the visual scenery that are caught by neither characters' dialogues nor music valence but, still, are essential for the story understanding. The voice-over was performed by professional Italian actor and recorded in a studio insulated from environmental noise and provided with professional hardware (Neumann U87 ai microphone, Universal Audio LA 610 mk2 preamplifier, Apogee Rosetta converter, Apple MacOS) and software (Logic Pro 10.4) equipment comprising a set of microphones and filters to manipulate sounds rendering. Then, the voice track was adequately combined with the movie original soundtracks and dialogues. Fade-in and fade-out effects were introduced to smooth the auditory content at the beginning and end of each run to better

manage the transitions among the subsequent segments of the film. Music and voice were finally remixed.

We faithfully transcribed the soundscape (for example, human voices, narrator voice-over, environmental and natural sounds) of the movie into subtitles. Subtitles were written in different styles and colours according to the speaking voice to facilitate speech segmentation and aid understanding. As line segmentation does not interfere with either reading and story comprehension or image processing[57], the subtitle pattern was modified in subsequent visual displays upon necessity, appearing in both two-line and one-line format. Video editing was carried out using iMovie software (10.1.10) on an Apple MacBook Pro, whereas for the creation of subtitles, we rely on the open-source cross-platform Aegisub 3.2.2 (http://www.aegisub.org/). In the visual and audiovisual conditions, a small red fixation cross was superimposed at the centre of the display, whereas subtitles were shown in the lower section of the screen.

## fMRI experimental design
Before starting the scanning session, participants were asked to rate their general knowledge of the movie plot on a Likert scale ranging from 1 (not at all) to 5 (very well).

Participants of each experimental sample were presented with one of the edited versions of the movie (visual, auditory or audiovisual) while undergoing fMRI recordings. Participants were instructed to simply enjoy the movie. Structural and functional data acquisition was performed on a single scanning day. After the scanning session, an ad hoc two-alternative forced-choice questionnaire about the content of the story was administered to assess subject engagement and compliance. In addition, other psychometric scales were administered to participants (Supplementary Information).

## Stimulation setup
Audio and visual stimulation were delivered through MR-compatible LCD goggles and headphones (VisualStim Resonance Technology, video resolution 800 × 600 at 60 Hz, visual field 30° × 22°, 5″, audio 30 dB noise attenuation, 40 Hz to 40 kHz frequency response). Both goggles and headphones were prescribed irrespectively of the experimental condition and group membership, meaning that each subject wore both devices. The video and audio clips were administered through software package Presentation 16.5 (Neurobehavioral System; http://www.neurobs.com).

## fMRI data acquisition and preprocessing
Brain activity was recorded with Philips 3T Ingenia scanner equipped with a 32-channel head coil. Functional images were acquired using gradient recall echo planar imaging (GRE-EPI; repetition time (TR) 2,000 ms; echo time (TE) 30 ms; flip angle (FA) 75°; field of view (FOV) 240 mm; acquisition matrix (in plane resolution) 80 × 80; acquisition slice thickness 3 mm; acquisition voxel size 3 × 3 × 3 mm; reconstruction voxel size 3 × 3 × 3 mm; 38 sequential axial ascending slices; total volumes 1,614 for the six runs of the movie, plus 256 for the control run). In the same session, three-dimensional high-resolution anatomical image of the brain was also acquired using a magnetization-prepared rapid gradient echo sequence (TR 7 ms; TE 3.2 ms; FA 9°; FOV 224, acquisition matrix 224 × 224; slice thickness 1 mm; voxel size 1 × 1 × 1 mm; 156 sagittal slices). Data collection and analysis were not performed blind to the conditions of the experiments.

fMRI data preprocessing was performed following the standard steps with AFNI_17.1.12 software package[58]. First, we removed scanner-related noise correcting the data by spike removal (3dDespike). Then, all volumes comprising a run were temporally aligned (3dTshift) and successively corrected for head motion using as base the first run (3dvolreg). A spatial smoothing with a Gaussian kernel (3dBlurToFWHM, 6 mm, full width at half maximum) was applied and then data of each run underwent percentage normalization.

In addition, detrending applying Savitzky–Golay filtering (sgolayfilt, polynomial order: 3, frame length: 200 timepoints) in MATLAB R2019b (MathWorks) was performed onto the normalized runs to smooth the corresponding time series and clean them from unwanted trends and outliers. Runs were then concatenated, and multiple regression analysis was performed (3dDeconvolve) to remove signals related to head motion parameters and movement spike regressors (framewise displacement above 0.3). Afterwards, single-subject fMRI volumes were non-linearly (3dQWarp) registered to the MNI-192 standard space[59].

## Computational modelling
We took advantage of computational modelling to extract a collection of movie-related features. Specifically, two sets of low-level features were defined, one extracted from the auditory stream (spectral[30] and sound envelope[60] properties to account for frequency- and amplitude-based modulations) and one from the visual stream (set of static Gabor-like filters -GIST[61,62] and motion energy information based on their spatiotemporal integration[63]). Moreover, high-level features were modelled on the basis of a manual tagging of natural and artificial categories occurring in both the auditory and visual streams, as well as using word embeddings from subtitles. As concerns the latter set of features, we built two alternative embeddings: (1) one from single sentences defined upon subtitling constraints, thus taking into full account semantic compositionality by means of the pretrained English-based GPT-3 algorithm[64] and (2) one using single word embeddings obtained through the Word2Vec algorithm[65] trained on an Italian corpus. Therefore, two high-level semantic models were defined, one combining categorical information and GPT-3 embeddings, and the other by combining categorical information and Word2Vec embeddings. As deep learning models, like GPT-3, have been proven to predict neural activity better than classical word embedding algorithms[32] (Supplementary Fig. 4), we included here as high-level model the one based on GPT-3 embeddings (and categorical semantics), whereas the results from Word2Vec embeddings were reported in Supplementary Fig. 5.

Finally, a set of features related to the movie editing process (for example, scene transitions, cuts, dialogues, music and audio descriptions) were manually annotated. A detailed description of these computational models as well as the parameters used to extract the features for the movie are reported in Supplementary Methods. The correspondences between sensory streams shown in Fig. 1c are described in Supplementary Methods.

To clean all stimulus models from a substantial portion of common variance, each model was orthogonalized from the movie editing descriptors. Indeed, the movie editing features (for example, film cuts, scene transitions, presence of dialogues, and music) have an impact both on the low-level (for example, transitions between scenes are often marked by a switch in the movie visual and auditory properties) and the high-level semantic descriptors (for example, spoken and written dialogues) and could, in principle, mask the impact of the fine-grained computational features, inflating the explained variance of each model. Thus, for every column of each model, a multiple regression was performed to predict computational features by using the movie editing descriptors as predictors. This procedure generated residuals from the features that became our computational models used in the encoding procedure and for the model-mediated ISC (both described below). Moreover, this procedure computed the portion of unique variance explained by each model discarding a large percentage of shared information (Supplementary Fig. 2b).

As model dimensionality was largely different across computational models (from a few columns of the acoustic model to hundreds in the visual one), an encoding procedure was performed in the multisensory AV condition only, to both verify the goodness of our descriptors and to prune irrelevant features. The methodological approach and results of this procedure are detailed in Supplementary Information.

## Across-condition ISC analysis

The ISC analysis was first performed in the group of subjects exposed to the multimodal (that is, audiovisual) stimulation. Therefore, for any given voxel included in our grey matter mask, the preprocessed time series of brain activity was extracted, and the average Pearson correlation coefficient ($r$) was calculated over every possible pair of subjects[9]. Moreover, an across-condition (A versus V) ISC analysis was computed through the synchronization of brain activity across individuals presented with a unimodal version of the movie. Notably, this procedure translates in evaluating, voxel by voxel, the correlation in the BOLD activity evoked in the subjects listening to the movie with that elicited in those watching the movie. Therefore, subject pairings were made across conditions matching individuals exposed to different sensory experiences.

To test the statistical significance of the ISC values, a non-parametric permutation test was run by generating surrogate voxel signals splicing the original data into 18 chunks (3 for each run) that were randomly re-arranged and eventually time-reversed (1,000 permutations). This procedure allowed to generate a null distribution that shared the same parameters (for example, mean and standard deviation) of the original data, as well as similar (but not identical) temporal dynamics. In detail, for each voxel, the correlation coefficient was evaluated over every possible pair of subjects, and from this set of coefficients a $t$-statistic was estimated (that is, dividing the mean $r$ coefficient by its standard error). The same procedure was also carried out with 1,000 surrogate voxel time series, thus obtaining a null distribution of $t$-statistics that provided the one-tailed $P$ value. $P$ values were estimated using a generalized Pareto distribution to the tail of the permutation distribution[66]. Correction for multiple comparisons was provided by thresholding statistical maps at the 95th percentile ($P < 0.05$, FWEc) of the maximum $t$ distribution from the permutation[67]. Finally, a conjunction analysis in TD participants between the AV and A versus V conditions was performed to evaluate the brain regions commonly synchronized across non-deprived groups. This conjunction analysis aimed at identifying and characterizing a core set of brain areas synchronized during the full multisensory experience and that equally shared a common representation across the unimodal processing of auditory and visual information in TD participants. The resulting map was subsequently used for the evaluation of the ISC in the A versus V condition in the deprived groups, as well as the TRWs and the model-mediated ISC across all experimental conditions.

## Model-mediated ISC

To assess the role of each model, an algorithm based on mediation analysis[68] was developed. The idea behind mediation analysis relies on the fact that a mediating factor intervenes in the relationship between the independent and the dependent variables. Here each model was used as a mediating factor during ISC. Specifically, before computing the ISC, the model contribution in the prediction of the BOLD signal was first removed in each subject separately through multiple regression. This procedure generated an ISC value for each model that represented the residual synchronization among subjects independent from our stimulus descriptors. For example, in a voxel that showed high ISC, a model able to predict most of its neural activity would generate a model-mediated ISC close to zero. Thus, the synchronization across subjects would critically depend on the features represented in that model. Conversely, a voxel showing high ISC before and after the mediation analysis would be interpretable as a voxel with an elevated synchronization across subjects, driven by unspecified (at least according to the considered models) activity. As concerns the low-level model, auditory and visual information were removed from the brain activity of those samples presented with the unimodal V and A condition, respectively, whereas the high-level semantic model, which was inherently multimodal, was removed from both conditions.

To obtain a statistical measure on the mediation effects, a permutation test was performed generating surrogate voxel signals splicing the original data similarly to the ISC described above. This allowed us to have a null distribution of model-mediated ISC values for each voxel and model. The statistical analysis was performed by evaluating the 'drop' (that is, ISC minus model-mediated ISC) and by comparing its intensity with the ones obtained from the null distribution. $P$ values were estimated using a generalized Pareto distribution to the tail of the permutation distribution[66] and corrected for multiple comparisons as above ($P < 0.05$, FWEc[67]). To evaluate the differences of the mediation effect between low- and high-level models, a two-sided Wilcoxon signed rank test was performed in each voxel by comparing the drop of ISC magnitude for the low-level model with the one produced by the high-level movie descriptor (Fig. 4e,f). Statistical threshold was defined through a permutation test (1,000 iterations) and correction for multiple comparisons was provided by thresholding statistical maps at the 95th percentile ($P < 0.05$, FWEc) of the distribution of maximum $W$ (that is, the sum of the ranks of positive differences between the two models) across permutations[67].

## TRW analysis

Sensory, perceptual and cognitive processes in the brain rely on the accumulation of information over different timespans. To measure the hierarchical organization of information processing, we computed the ISC over overlapping, binned segments of data. Specifically, during ISC, the Pearson correlation coefficient was calculated after averaging consecutive timepoints over overlapping rectangular sliding windows ranging from 2 s to 240 s and moving in steps of 2 s. Subsequently, for each voxel, we extracted the width of the window showing the highest synchronization (that is, highest correlation coefficient) across subjects, named as TRW, which is conceptually similar to the approach of Hasson and colleagues[10,40]. To test the statistical significance of the TRW, similarly to ISC, a non-parametric permutation test was performed by evaluating correlation coefficients using surrogate voxel time series (1,000 permutations) using the specific temporal tuning of each voxel. $P$ values were estimated using a generalized Pareto distribution to the tail of the permutation distribution[66] and corrected for multiple comparisons as above ($P < 0.05$, FWEc[67]).

Similarity of the TRW maps across the three experimental conditions was evaluated in the subset of voxels which resulted to be significant in the A versus V condition in SD participants. Specifically, we first estimated Spearman's $\rho$ across maps (for example, between TD and SD for the A versus V condition). To test the statistical significance of these similarities, for each experimental condition, we generated 200 null TRW maps by means of surrogate signals, as described above. This latter procedure resulted in a set of maps with the same spatial smoothness of the original data. $P$ values were estimated by comparing the actual $\rho$ with the ones obtained from the 200 null maps ($P < 0.005$).

## Reporting summary

Further information on research design is available in the Nature Portfolio Reporting Summary linked to this article.

# Data availability

fMRI data are available on https://osf.io/j8x6h/. Only preprocessed functional data were shared. Raw structural and functional MRI data are available from the corresponding author upon reasonable request to comply with the European General Data Protection Regulation (GDPR). Cortical parcellation was performed using the Human Connectome Project Atlas[11], projected onto the MNI template (https://identifiers.org/neurovault.collection:1549).

# Code availability

Code is available here: https://github.com/giacomohandjaras/101_Dalmatians and also at https://osf.io/j8x6h/ website.

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

## Acknowledgements

This work has been supported by a PRIN grant (2017_55TKFE) by the Italian Ministry of University and Research to P.P. Additionally, F.S. was supported by the Frontier Proposal Fellowship (FPF program, 2019) granted by IMT School for Advanced Studies Lucca; M.D. was supported by the ERC Consolidator Grant LIGHTUP (project #772953). The funders had no role in study design, data collection and analysis, decision to publish or preparation of the manuscript. We thank the Unione Italiana Ciechi e Ipovedenti (The Italian Union of the Blind and Partially Sighted) and the Ente Nazionale Sordi Onlus (The Italian Union of the Deaf) for their support. A preliminary version of this manuscript has undergone an internal review by L. Merabet. We acknowledge V. Cardin and O. Collignon for insightful discussions on the project.

## Author contributions

F.S., G.H., D.B., A.L., P.P. and E.R., conceptualization; F.S., A.L. and G.H., methodology, software and formal analysis; F.S., A.L. and M.D, investigation; F.S., G.H., F.G., V.B. and C.T., resources; F.S. and G.H., data curation; F.S., G.H., D.B., L.C., F.G., P.P. and E.R., interpreted results of experiments; F.S., G.H., D.B. and E.R., writing—original draft; all authors, writing—review and editing; F.S. and G.H., visualization; F.G., D.B., L.C., P.P. and E.R., supervision; F.G., P.P. and E.R., project administration; F.G., P.P. and E.R., funding acquisition.

## Competing interests

The authors declare no competing interests.

## Additional information

**Correspondence and requests for materials** should be addressed to Emiliano Ricciardi.

# Reporting Summary

## Statistics

For all statistical analyses, confirm that the following items are present in the figure legend, table legend, main text, or Methods section.

| n/a | Confirmed | |
|---|---|---|
| ☐ | ☒ | The exact sample size (n) for each experimental group/condition, given as a discrete number and unit of measurement |
| ☐ | ☒ | A statement on whether measurements were taken from distinct samples or whether the same sample was measured repeatedly |
| ☐ | ☒ | The statistical test(s) used AND whether they are one- or two-sided *Only common tests should be described solely by name; describe more complex techniques in the Methods section.* |
| ☐ | ☒ | A description of all covariates tested |
| ☐ | ☒ | A description of any assumptions or corrections, such as tests of normality and adjustment for multiple comparisons |
| ☐ | ☒ | A full description of the statistical parameters including central tendency (e.g. means) or other basic estimates (e.g. regression coefficient) AND variation (e.g. standard deviation) or associated estimates of uncertainty (e.g. confidence intervals) |
| ☐ | ☒ | For null hypothesis testing, the test statistic (e.g. $F$, $t$, $r$) with confidence intervals, effect sizes, degrees of freedom and $P$ value noted *Give P values as exact values whenever suitable.* |
| ☒ | ☐ | For Bayesian analysis, information on the choice of priors and Markov chain Monte Carlo settings |
| ☒ | ☐ | For hierarchical and complex designs, identification of the appropriate level for tests and full reporting of outcomes |
| ☐ | ☒ | Estimates of effect sizes (e.g. Cohen's $d$, Pearson's $r$), indicating how they were calculated |

*Our web collection on statistics for biologists contains articles on many of the points above.*

## Software and code

Policy information about availability of computer code

| Data collection | Sound mixing was performed with the software from Apple® LogicPro 10.4. The video and audio clips were edited with iMovie software from Apple® (10.1.10) whereas for the creation of subtitles, we rely on the open-source cross-platform Aegisub 3.2.2 (http://www.aegisub.org/). Stimulation was administered through software package Presentation® 16.5 (Neurobehavioral System, Berkeley, CA, USA - http://www.neurobs.com). Brain activity was recorded with Philips 3T Ingenia scanner equipped with a 32-channel head coil. |
|---|---|
| Data analysis | fMRI data preprocessing and analysis was performed following the standard steps with AFNI_17.1.12 software package and MATLAB R2019b (MathWorks Inc., Natick, MA, USA). Code is available at https://github.com/giacomohandjaras/101_Dalmatians |

For manuscripts utilizing custom algorithms or software that are central to the research but not yet described in published literature, software must be made available to editors and reviewers. We strongly encourage code deposition in a community repository (e.g. GitHub). See the Nature Portfolio guidelines for submitting code & software for further information.

## Data

Policy information about availability of data

All manuscripts must include a data availability statement. This statement should provide the following information, where applicable:
- Accession codes, unique identifiers, or web links for publicly available datasets
- A description of any restrictions on data availability
- For clinical datasets or third party data, please ensure that the statement adheres to our policy

fMRI data are available on https://osf.io/j8x6h/. Only preprocessed functional data was shared. Raw structural and functional MRI data are available from the corresponding author upon reasonable request to comply with the European General Data Protection Regulation (GDPR). Cortical parcellation was performed using

the HCP Atlas (Glasser et al., 2016, https://www.nature.com/articles/nature18933), projected onto the MNI template (https://identifiers.org/neurovault.collection:1549).

# Field-specific reporting

Please select the one below that is the best fit for your research. If you are not sure, read the appropriate sections before making your selection.

☐ Life sciences   ☒ Behavioural & social sciences   ☐ Ecological, evolutionary & environmental sciences

For a reference copy of the document with all sections, see nature.com/documents/nr-reporting-summary-flat.pdf

# Behavioural & social sciences study design

All studies must disclose on these points even when the disclosure is negative.

| | |
|---|---|
| Study description | 3T fMRI study with a naturalistic paradigm (i.e., a movie) administered via three distinct experimental conditions: i) audiovisual, ii) visual-only and iii) auditory-only stimulations. Quantitative, cross-sectional data were acquired. |
| Research sample | Fifty subjects took part in the study. We enrolled both typically developed (TD) individuals and sensory deprived (SD) subjects, who lack visual or auditory experience since birth. Three samples of TD individuals underwent a different experimental condition consisting in the presentation of one version of the same movie: either i) the full multimodal audiovisual (AV) (n=10, 35±13 years, 8 females), ii) the auditory (A) (n=10, 39±17 years, 7 females) or iii) the visual (V) (n=10, 37±15 years, 5 females) one. SD individuals comprising blind (n=11, mean age 46±14 years, 3 females) and deaf (n=9, mean age 24±4, 5 females) participants were presented with the A and V movie conditions respectively. Congenitally blind and deaf subjects in Europe are extremely rare (<4 out of 10000 newborns for blindness and <2 out of 1000 births for deafness). Moreover, additional eligibility criteria (no metal implants, no medication, no history of neurological or psychiatric disorders, all native Italian speakers) constrained samples recruitment. Sample sizes are comparable to other studies in the field of sensory deprivation. Since our aim was to test the extent to which audiovisual experience is a mandatory prerequisite for the superior temporal cortex to develop and become able to detect shared features between the two sensory streams, we enrolled adult individuals who specifically lack visual or auditory input since birth. |
| Sampling strategy | The calculation of a suitable sample size was determined by a review of the literature of fMRI studies with naturalistic stimulation and using Intersubject Correlation (ISC) analysis. Results of this procedure are presented in Supplementary Fig.6. Considering the duration of naturalistic stimulation and the subejcts enrolled in the reviewed studies, our sample sizes match with those reported in the literature. Since congenitally deafness and blindness are extremely rare conditions, subject recrtuitment was performed within the two major Italian organizations: the Unione Italiana Ciechi e Ipovedenti (Italian Union of the Blind and Partially Sighted) and the Ente Nazionale Sordi Onlus (Italian Union of the Deaf). Sighted and normal hearing individuals were recruited by word of mouth. |
| Data collection | Structural and functional data acquisition were performed on a single scanning day during the presentation of the narrative. Brain activity was recorded with Philips 3T Ingenia scanner equipped with a 32-channel head coil. Functional images were acquired using gradient recall echo planar imaging (GRE-EPI). In the same session, three-dimensional high-resolution anatomical image of the brain was also acquired using a magnetization-prepared rapid gradient echo (MPRAGE). Audio and visual stimulation were delivered through MR-compatible LCD goggles and headphones (VisualStim Resonance Technology, video resolution 800x600 at 60 Hz, visual field 30° × 22°, 5, audio 30 dB noise-attenuation, 40 Hz to 40 kHz frequency response). Both goggles and headphones were prescribed irrespectively of the experimental condition and group membership, meaning that each subject wore both devices. The video and audio clips were administered through software package Presentation® 16.5 (Neurobehavioral System, Berkeley, CA, USA - http://www.neurobs.com). Congenitally deaf individuals were accompanied by a sign language interpreter who explained the task and assisted them. Researchers were not blind to the experimental conditions. |
| Timing | Data acquisition started in December 2018 and ended in October 2019. |
| Data exclusions | Two blind subjects were removed from the fMRI analysis for excessive head movement (final sample, n=9, mean age 44±14 years, 3 females). |
| Non-participation | No participants dropped out or declined the participation. |
| Randomization | TD participants were randomly allocated to the audiovisual, visual-only or auditory-only conditions. |

# Reporting for specific materials, systems and methods

We require information from authors about some types of materials, experimental systems and methods used in many studies. Here, indicate whether each material, system or method listed is relevant to your study. If you are not sure if a list item applies to your research, read the appropriate section before selecting a response.

## Materials & experimental systems

| n/a | Involved in the study |
|-----|----------------------|
| ☒ ☐ | Antibodies |
| ☒ ☐ | Eukaryotic cell lines |
| ☒ ☐ | Palaeontology and archaeology |
| ☒ ☐ | Animals and other organisms |
| ☐ ☒ | Human research participants |
| ☒ ☐ | Clinical data |
| ☒ ☐ | Dual use research of concern |

## Methods

| n/a | Involved in the study |
|-----|----------------------|
| ☒ ☐ | ChIP-seq |
| ☒ ☐ | Flow cytometry |
| ☐ ☒ | MRI-based neuroimaging |

# Human research participants

Policy information about studies involving human research participants

| | |
|---|---|
| Population characteristics | Causes of congenital blindness were heterogenous: retinopathy of prematurity, retinitis pigmentosa, optic nerve atrophy, Leber congenital amaurosis, retinal detachment, bilateral retinoblastoma. All blind participants did not have any residual light perception. They all learned Braille reading at the age of six. All deaf participants reported to suffer from a hereditary form of congenital deafness but one subject who underwent sensorineural hearing loss in the first months of life due to high fever. They reported to have learned Italian Sign Language (LIS) as the first language. Eight out of nine congenitally deaf participants used hearing aids during childhood, while one of them still utilized the device in his life at the moment of the study. For additional demographic characteristics (e.g., age, sex) of the samples please see above. |
| Recruitment | Due to the extremely low prevalence of congenitally deaf and blind individuals in the population, sensory deprived participants were recruited using the snowball sampling procedure. Typically developed individuals were recruited within the students of the Psychology program at the University of Turin as well as by word of mount among their acquaintances to match age and sex characteristics. |
| Ethics oversight | The study was approved by the Ethical Committee of the University of Turin (protocol n. 195874, 05/29/19) and conforms to the Declaration of Helsinki. |

Note that full information on the approval of the study protocol must also be provided in the manuscript.

# Magnetic resonance imaging

## Experimental design

| | |
|---|---|
| Design type | Naturalistic Stimulation |
| Design specifications | Movie presentation was split in 6 runs of about 8 minutes duration each. Afterwards, the experimental paradigms involved an additional (8 minutes) run consisting in a scrambled version of the narrative, as control condition for the processing of the story semantic. Runs were presented consecutively and were separated by brief intervals in which the experimenter communicated with the subjects to check their compliance and ascertain everything was good during the scanning. No specific task was provided to the participants that were just told to "follow the plot and enjoy the movie". |
| Behavioral performance measures | Before starting the scanning session, participants were asked to rate their general knowledge of the movie plot on a Likert scale ranging from 1 (not at all) to 5 (very well), and at the end of the experiment, an ad hoc two-alternative forced choice questionnaire about the content of the story was administered at the end of the experiment to assess subject engagement and compliance. |

## Acquisition

| | |
|---|---|
| Imaging type(s) | Structural and functional MRI data. |
| Field strength | 3T |
| Sequence & imaging parameters | Functional images: GRE-EPI; TR = 2000 ms; TE = 30 ms; FA = 75°; FOV = 240 mm; matrix size (in plane resolution) = 80 × 80; slice thickness = 3 mm; voxel size =3x3x3 mm; 38 sequential axial ascending slices. Anatomical image: MPRAGE; TR =7 ms; TE = 3.2 ms; FA = 9°; FOV= 224, matrix size = 224 x 224; slice thickness = 1mm; voxel size = 1x1x1 mm; 156 sagittal slices). |
| Area of acquisition | whole brain scans |
| Diffusion MRI | ☐ Used    ☒ Not used |

## Preprocessing

| | |
|---|---|
| Preprocessing software | AFNI_17.1.12 and MATLAB R2019b (MathWorks Inc., Natick, MA, USA) software packages were used. |

| Normalization | single subject fMRI volumes were nonlinearly (3dQWarp) registered to the MNI-192 standard space. |
|---|---|
| Normalization template | MNI-192 standard space. |
| Noise and artifact removal | we removed scanner-related noise correcting the data by spike removal (3dDespike). Head motion was corrected using as base the first run (3dvolreg). A multiple regression analysis was performed (3dDeconvolve) to remove signals related to head motion parameters, and movement spike regressors (frame wise displacement above 0.3). Signal trends were removed using a Savitzky-Golay filter in Matlab. |
| Volume censoring | no volumes were excluded from the analysis. |

## Statistical modeling & inference

| Model type and settings | Inter-Subject Correlation (ISC) during a naturalistic stimulation, considering subject pairings as a random effect. Statistical significance of the inter-subject synchronization was evaluated throughout permutation tests by shuffling fMRI time series preserving low- and high-frequency fluctuations. A model-mediated version of ISC was developed to measure the impact on subjects' synchronization of multiple sets of computational features. Statistical significance of the model-mediated ISC was obtained using null models generated by means of the IAAFT (Iterative Amplitude Adjusted Fourier Transform) algorithm, to preserve spectral as well as distributional properties of the original features. |
|---|---|
| Effect(s) tested | ISC during multimodal stimulation (average Pearson correlation coefficient of every possible subjects' pair in each voxel independently). ISC across modality, in which we measured the synchronization between subjects exposed to the auditory stimulation only and those administered with the visual stimulation only. ISC across modality was performed in typically developed (TD) individuals and sensory deprived (SD) subjects. We tested in each condition whether ISC was significant greater than zero (one-tailed, non-parametric test), and the differences of ISC magnitude between TD and SD participants (two-tailed, Wilcoxon rank sum test). As concerns the model-mediated ISC, we tested whether each set of computational features was significantly able to reduce the across-modality synchronization in TD and SD participants (one-tailed non-parametric test). |

Specify type of analysis: ☐ Whole brain  ☐ ROI-based  ☒ Both

| Anatomical location(s) | *Describe how anatomical locations were determined (e.g. specify whether automated labeling algorithms or probabilistic atlases were used).* |
|---|---|
| Statistic type for inference (See Eklund et al. 2016) | voxel-wise |
| Correction | FWE, Bonferroni. |

## Models & analysis

| n/a | Involved in the study |
|---|---|
| ☒ ☐ | Functional and/or effective connectivity |
| ☒ ☐ | Graph analysis |
| ☒ ☐ | Multivariate modeling or predictive analysis |

