## [Peer Review File · Nature Human Behaviour]

Peer Review Information

Journal: Nature Human Behaviour

Manuscript Title:

Corresponding author name(s):

Reviewer Comments & Decisions:

Decision Letter, initial version:

19th May 2022

Dear Professor Ricciardi,

Thank you once again for your manuscript, entitled "A modality independent proto-organization of human multisensory areas", and for your patience during the peer review process.

Your Article has now been evaluated by 2 referees. You will see from their comments copied below that, although they find your work of potential interest, they have raised quite substantial concerns. In light of these comments, we cannot accept the manuscript for publication, but would be interested in considering a revised version if you are willing and able to fully address reviewer and editorial concerns.

We hope you will find the referees' comments useful as you decide how to proceed. If you wish to submit a substantially revised manuscript, please bear in mind that we will be reluctant to approach the referees again in the absence of major revisions. We are committed to providing a fair and constructive peer-review process. Do not hesitate to contact us if there are specific requests from the reviewers that you believe are technically impossible or unlikely to yield a meaningful outcome.

In particular, as well as addressing all of the other reviewer points, it will be important to fully address the statistical points raised by Reviewer #2 in their final paragraph.

If you wish to submit a suitably revised manuscript we would hope to receive it within 4 months. I

would be grateful if you could contact us as soon as possible if you foresee difficulties with meeting this target resubmission date.

- Include a "Response to the editors and reviewers" document detailing, point-by-point, how you addressed each editor and referee comment. If no action was taken to address a point, you must provide a compelling argument. When formatting this document, please respond to each reviewer comment individually, including the full text of the reviewer comment verbatim followed by your response to the individual point. This response will be used by the editors to evaluate your revision and sent back to the reviewers along with the revised manuscript.
- Highlight all changes made to your manuscript or provide us with a version that tracks changes.

[REDACTED]

Thank you for the opportunity to review your work. Please do not hesitate to contact me if you have any questions or would like to discuss the required revisions further.

Sincerely,
Jamie

Dr Jamie Horder
Senior Editor
Nature Human Behaviour

REVIEWER COMMENTS:

Reviewer #1:
Remarks to the Author:

In this article, the authors analyze fMRI responses to auditory, visual, and audiovisual narratives in typically developing participants, as well as responses to auditory narratives in congenitally blind participants, and responses to visual narratives in congenitally deaf participants. Using intersubject correlations (ISC), they report a variety of findings including the key results that 1) significant ISC values can be detected between SD participants across modalities, 2) regressing out low-lever perceptual features reduces cross-modal ISCs, and 3) regions showing cross-modal ISC show differential sensitivity to information at different timescales. The article investigates an important question using a sophisticated combination of techniques. The following points will need to be

addressed to further improve the article:

1. The study focuses on the presence of significant ISC across the SDs. This is a very interesting finding, but another key finding is that in several regions ISC across modalities was greater in TD participants than in SD participants (Figure 3B). In addition, there is a marked difference in the temporal receptive windows A vs V between TD and SD (Figure 5). What does that say about the role of multisensory experience for shaping multisensory representations?

2. There is a difference between co-occurrence of auditory and visual low level information in the same regions vs integration of multisensory information. In phenomena such as the McGurk effect, auditory and visual information do not merely coexist in the same brain regions, but they interact such that changes to the stimuli in one modality can affect the perception of the other modality. My understanding is that while the study is already very interesting and informative, it remains unknown whether such integration mechanisms are part of the "proto-organization" or whether they require multisensory experience. It would be important to discuss this in the discussion section.

3. Word2vec was used as a model of semantics, but recent work (Schrimpf et al., see reference below) indicates that other models provide a much better fit of neural responses. This limitation should be acknowledged in the discussion, and the potential impact of this on the results should be discussed.

Schrimpf, M., Blank, I., Tuckute, G., Kauf, C., Hosseini, E. A., Kanwisher, N., ... & Fedorenko, E. (2020). The neural architecture of language: Integrative reverse-engineering converges on a model for predictive processing. bioRxiv.

4. The author claim that "the overall TRW profile was consistent across the three experimental procedures, indicating that synchronized regions retained a similar chronotopic organization". This conclusion should be supported by a quantitative analysis, e.g. by correlating the TRW maps across the three procedures.

5. "The present study questioned whether brain regions involved in audiovisual processing, including the superior temporal and neighboring areas, retain the ability to represent sensory correspondences across modalities, despite the complete lack of any prior auditory or visual experience."

I believe the authors mean "The present study asked..." or "The present study tested...". The expression "to question something" means to feel or express doubt about that thing, or to raise objections about it.

6. In the discussion: "Synchronized neural responses, selectivity for perceptual features and a preserved mapping of temporal receptive windows demonstrate that the functional architecture of the superior temporal cortex emerges despite the lack of audiovisual inputs since birth and irrespectively of postnatal sensory experiences."

I believe that this claim is too strong and not supported by the results. The way this sentence is phrased, it seems that the authors are saying that all aspects of the functional architecture of superior temporal cortex are independent of experience. However, this cannot be tested with existing methods, because the resolution of functional MRI is not adequate to measure all aspects of the architecture of superior temporal cortex. In addition, as already noted in point 3, the results also reveal substantial

differences between the responses in TD and SD participants. The findings are already very interesting without the need of making claims that are not supported by the data. I suggest that the authors moderate this claim and discuss the fact that limits of the instruments mean that many aspects of the functional architecture might not be observable with the approach used.

7. In the discussion, this sentence "Conversely, the refinement of more complex levels of audiovisual skills appears to require a full multisensory experience throughout lifetime." does not seem to be directly supported by the results in the article, therefore it needs to be supported by references to the literature. Alternatively, it should be phrased as a hypothesis or speculation rather than as a statement of fact.

8. "Here, we observed that the high-level movie properties, such as word embeddings and categorical semantic information, mediate the synchronization across auditory and visual stimulations in TD but not in the SD participants. While perceptual features are intrinsic properties of a given stimulus, higher order characteristics rely more on experience and learning, and this may partially account for the observed differences between TD and SD participants."

This is a surprising result, as categorical semantic information should be the least dependent on changes in perceptual modality. Indeed, previous work (Bedny et al. 2019, see reference below) shows that sensory deprived participants still retain largely intact semantic knowledge – this might well depend on experience and learning, but it appears that even SD participants would have access to the necessary experience (e.g. auditory experience alone might be sufficient to acquire the semantic knowledge). Please reference the relevant previous work and discuss how you might reconcile your results with the previous findings.

Bedny, M., Koster-Hale, J., Elli, G., Yazzolino, L., & Saxe, R. (2019). There's more to "sparkle" than meets the eye: Knowledge of vision and light verbs among congenitally blind and sighted individuals. *Cognition*, 189, 105-115.

9. In Figures 2a, 3a, 4 and 5, please include the extreme values for both ends of the color bars (only one extreme is provided).

Stefano Anzellotti

Reviewer #2:
Remarks to the Author:

In this manuscript, Setti and colleagues present a modified version of the Disney movie "101 Dalmations" (or "Centouno Dalmata" in Italian) and measure the resultant brain activity using fMRI. The movie was presented in either auditory (A), visual (V) or audiovisual (AV) formats. Because the A and V movies are missing information relative to the AV movies, they were supplemented with subtitles for the V movie and with a voice-over describing the missing visual information for the A movie. The movie was presented to separate groups of 10 typically-developed (TD) individuals in A, V,

and AV formats; to 10 blind individuals in the A format; and to 10 deaf individuals in the V format (deaf and blind individuals are described as sensory-deprived, SD).

The analysis is based on the idea of intersubject correlation (ISC), measuring whether brain regions that show similar fMRI time courses across participants. ISC can be considered a hybrid between resting-state fMRI analysis methods, which analyze correlations between the fMRI time course of functionally connected regions within a single subject; and task-based fMRI analysis methods, which analyze correlations between an externally-presented stimulus event and the fMRI course of a region. ISC is well-suited to analyzing the brain response to lengthy or naturalistic stimuli such as movies. The main result from the manuscript is that ISC is found across a good fraction of the brain, especially superior temporal cortex, lateral frontal cortex, and "default mode network" areas. This is true even for movies presented in different modalities (A vs. V) across TD or SD individuals. The manuscript has many good qualities. The effort necessary to scan both deaf and blind SD individuals clearly sets it apart from most fMRI studies. The ISC analysis is sensible and the finding that temporal cortex synchronizes to movies presented in different modalities is clearly interesting.

The manuscript also has some flaws. The big claim about superior temporal cortex (STC) containing "a modality-independent topographical organization of temporal dynamics" seems only weakly supported by the data. One reason is that fMRI is not an ideal tool for measuring temporal windows. For instance, Luo and Poeppel (2012) use MEG data as evidence that STC contains two temporal windows, one of 20-80 ms, the other 150-300 ms. Given that these window durations are about 100-times faster than the duration of the hemodynamic response measured with BOLD (~15 seconds), it is unlikely that BOLD fMRI can map (or even distinguish) their topographical organization. This is not to say that slow signals in fMRI do not carry information, just that it is a fraught endeavor to tie them a specific neurophysiological property such as the duration of temporal integration windows in STC. As shown in Figure 5, most voxels have a window of ~15 seconds, exactly as you would expect if the window was determined by the autocorrelation induced by the slow BOLD temporal window, rather than any properties of the underlying neurons. In any event, to show that there is a topographical organization, it would be necessary to show a regular organization of window durations within a small region of cortex (akin to a tonotopic map) and the authors do not show any such data. Instead, Figure 5 shows that some areas are measured to have shorter windows, and others are measured to have longer, but this is not equivalent to a topographical organization.

One disadvantage of ISC is that it can be difficult to know what exactly in the movie is triggering the observed correlation between subjects. The properties of the movie can be examined in an attempt to explain observed correlations. For instance, in the original Hasson et al. paper, movie segments containing faces evoked strong ISC in the fusiform face area (presumably since all subjects' FFAs showed strong responses at these times). Setti and colleagues use a quantitative modeling approach to measure the effect of low-level visual features, low-level acoustic features, and high-level semantic features, calculated from the stimulus movie.

In general, this approach is sensible and sophisticated, modeling low-level sensory features and high-level semantic features. However, it is not exactly clear how well the models worked. Figure S1 shows some data on how well the models predicted the observed MR time series. In the referenced study of Nishimoto et al., the correlation between predicted and observed BOLD fMRI was as high as $r = 0.40$. In the current study, it seems that the values are quite a bit lower. It would be good to compare the model fit values between the current study and previous studies that used similar

methods (many studies from Gallant and colleagues). It is also difficult to interpret the "model-mediated ISC", that is, the drop in ISC, which seems quite low (1-2%). It is not good to draw firm conclusions based on this small change and dichotomize areas as being responsive (or not) to the features in a model based on a tiny drop (or not) in ISC correlation, due to the statistical fallacy that the difference between "significant" (e.g. $p = 0.04$) and "not significant" (e.g. $p = 0.06$) is often not itself significant. The preferred method would be to perform a direct statistical comparison between the goodness-of-fit (between the modeled and measured fMRI time series) for models with and without a given feature included (the method used by Gallant and colleagues to compare different models).

Minor notes

line 757: "outern"?

Author Rebuttal to Initial comments

Response to the Editors and Reviewers Comments

Reviewer #1:

In this article, the authors analyze fMRI responses to auditory, visual, and audiovisual narratives in typically developing participants, as well as responses to auditory narratives in congenitally blind participants, and responses to visual narratives in congenitally deaf participants. Using intersubject correlations (ISC), they report a variety of findings including the key results that 1) significant ISC values can be detected between SD participants across modalities, 2) regressing out low-level perceptual features reduces cross-modal ISCs, and 3) regions showing cross-modal ISC show differential sensitivity to information at different timescales. The article investigates an important question using a sophisticated combination of techniques.

We thank the Reviewer for acknowledging the relevance of the study aim and the complexity of the methodologies used. We really appreciated all comments and suggestions that contributed to the improvement of our work. We addressed all issues that have been raised and we revised the manuscript accordingly.

The following points will need to be addressed to further improve the article:

Issue 1. The study focuses on the presence of significant ISC across the SDs. This is a very interesting finding, but another key finding is that in several regions ISC across modalities was greater in TD participants than in SD participants (Figure 3B). In addition, there is a marked difference in the temporal receptive windows A vs V between TD and

SD (Figure 5). What does that say about the role of multisensory experience for shaping multisensory representations?

Our study observed that a core set of regions - including bilateral temporal, temporo-parieto-occipital, parieto-occipital and posterior cingulate cortex - are synchronized across Sensory Deprived (SD) participants when processing coherent auditory and visual information, revealing their ability to represent sensory correspondences across modalities despite the lack of an integrated audio-visual experience since birth.

Furthermore, as shown in Fig.3 and discussed in the original main manuscript, several brain regions present differences in the ISC magnitude between Typically Developed (TD) and SD participants. Specifically, the comparison of the ISC between the two groups in the A vs V condition shows significantly greater values for the TD than SD participants in all the regions of the conjunction map, with the only exception of the posterior parietal cortex. Additionally, in the left inferior frontal and bilateral medial prefrontal cortex of the SD group, the ISC was not even significant.

These variations in the ISC magnitude in the SD groups likely result from distinct postnatal experiences and developmental trajectories that may affect the synchronization across blind and deaf individuals, and consequently between SD and TD individuals. Indeed, modifications in language processing exist between deaf and blind individuals (e.g., Trettenbrein et al., 2021; Neville et al., 1998; Pant et al., 2020; Handjaras et al., 2016). Moreover, differences in the naturalistic information processing may emerge as a result of diverse visually-only and acoustically-only semantic representation of events (see also Response 2). At the same time, in TD individuals, it is plausible that both learned associations and imagery in the other modality (i.e., the visual image of a dog while listening to its barking and vice versa) would enhance synchronization, even when processing unimodal information.

Although the ISC magnitude in the SD group was significantly lower than the one measured within the TD group, the residual shared synchronization suggests that the ability of STS/STG and neighboring areas to process signals originating from the same events in a modality-independent manner is retained despite the lack of a specific sensory experience (Fig.3a). On the other hand, the difference between TD and SD favors the hypothesis that a full functional refinement of audiovisual processing may require the typical interaction between these two senses during development and throughout lifetime.

As far as the similarities and differences in the temporal receptive windows (TRW) are concerned, please refer to Response 4 and to Response 1 to Reviewer 2, below.

Differences in ISC between TD and SD groups have been now examined both in the Supplementary Discussion section (page 3-4) and were further expanded in the Discussion of the revised manuscript (page 11 and 13).

Issue 2. There is a difference between co-occurrence of auditory and visual low level information in the same regions vs integration of multisensory information. In phenomena such as the McGurk effect, auditory and visual information do not merely coexist in the same brain regions, but they interact such that changes to the stimuli in one modality can affect the perception of the other modality. My understanding is that while the study is already very interesting and informative, it remains unknown whether such integration mechanisms are part of the “proto-organization” or whether they require multisensory experience. It would be important to discuss this in the discussion section.

Our study sought to determine whether the cortical network involved in processing audiovisual input is tuned to represent correspondences across sensory modalities, independently from any prior audiovisual experience. To this purpose, we characterized the coherence of neural responses to a naturalistic stimulation across modalities and across independent samples of TD and congenitally SD individuals. Any evidence of synchronization across conditions and groups has been considered as indicative of a shared representation of visual and auditory features despite and independently of different postnatal sensory experiences. These observations favor the hypothesis that the multisensory network may be naturally endowed with a functional scaffolding (i.e., proto-organization) to yield a common neural representation across coherent auditory and visual input.

Nonetheless, by definition, the effect of a multisensory stimulation and of the interaction among inputs (e.g., as in the McGurk effect) could not be assessed in individuals who lack either vision or hearing since birth. We completely agree with the Reviewer that measuring the synchronization of the neural response across two modalities does not allow claiming that we are investigating multisensory processing/integration. Therefore, we purposely avoided the potentially misleading usage of the term “multisensory integration” throughout the manuscript.

At the same time, our study does not aim to directly assess (nor could it) either the functionality of this ‘proto-organization’ toward multisensory integration, or the role of multisensory experience on the development of multisensory functions.

In this regard, evidence exists that basic multisensory functions are already present at birth. Despite perceptually inexperienced, newborns exploit low-level, sensory-independent statistical attributes (e.g., intensity, duration, tempo, rhythm, and spatial location) to combine stimuli across modalities (Gibson et al., 1970; Chandrasekaran et al., 2009; Munhall &

Vatikiotis-Bateson, 2004) and exhibit rudimentary multisensory functions (e.g., Kuhl & Meltzoff, 1982; Patterson & Werker, 2003; Walton & Bower, 1993; Lewkowicz et al., 2010; Lewkowicz and Turkewitz 1980). At the same time, experience during development appears to be crucial for the emergence of comprehensive multisensory representations (e.g., speech perception and semantic congruence) which involve multiple cognitive functions (e.g., object identification, language, memory, attention) (Murray et al., 2016; Stein et al., 2014; Lewkowicz, 2000; Hillock-Dunn and Wallace, 2012; Zhou et al., 2020).

Consistently, people who underwent visual or auditory restoration after an early, but temporary, lack of audiovisual experience offer another perspective to study the developmental constraints of multisensory functions. In fact, the short-term absence of a patterned visual input, and thus of a coherent audiovisual experience, in congenital cataract-reversal individuals did not affect basic audiovisual processes (Putzar et al., 2012), while hampering more complex multisensory functions (i.e., speech in noise -Putzar et al., 2007, Guerreiro et al., 2015). Similarly, an efficient audiovisual processing for the simple detection of events was found in individuals who experienced an early period of auditory deprivation and whose hearing was restored with cochlear implants (e.g., Gilley et al., 2010), while reduced multisensory gains were documented for tasks requiring the integration of complex audiovisual information (Desai et al. 2008; Huyse et al. 2013; Rouger et al. 2008; Schorr et al. 2005; Stropahl et al. 2015b; Tremblay et al. 2010; but see Tona et al. 2015).

Altogether, the above observations suggest that basic multisensory functioning might not require early audiovisual experience and the brain might be endowed with an intrinsic functional organization to combine coherent low-level properties of the incoming sensory information. On the other hand, a complete audiovisual experience appears to be fundamental for the full development of higher-level processes, which do not display a high degree of maturation at birth.

Compared to the literature in the field, we believe that the paradigm adopted here with the inclusion of two different models of sensory deprivation (i.e., congenitally blind, and deaf individuals) is an optimal experimental context to test whether the functional properties of the classically-defined multisensory cortex do require audiovisual experience to develop. Newborns and sensory-restored individuals represent alternative models to investigate the developmental constraints of sensory interplay, but the information that can be derived from studies exploiting these models is not conclusive. As a matter of fact, only basic sensory processes can be actually explored in newborns. Moreover, sensory processing is typically affected in individuals who experienced the restoration of a sensory input both by adaptive plasticity phenomena and degraded processing of the restored modalities. Our study

addresses the limitations discussed above by investigating the interplay across visual and auditory sensory streams in individuals who lack audiovisual experience since birth.

Following the Reviewer's comments, the Discussion of the revised manuscript (page 15) has been further expanded in light of the aforementioned observations and our statements on the role of multisensory experience restated.

Issue 3. Word2vec was used as a model of semantics, but recent work (Schrimpf et al., see reference below) indicates that other models provide a much better fit of neural responses. This limitation should be acknowledged in the discussion, and the potential impact of this on the results should be discussed.

As the Reviewer pointed out, other more recent computational algorithms can successfully model linguistic features. The choice of Word2Vec algorithm to model semantics was determined by several considerations about the properties of our naturalistic stimulus.

First, since our participants were native Italian speakers, we used the Italian version of the movie "101 Dalmatians" ("La carica dei 101"). Therefore, to extract semantic features, we used an Italian pre-trained set of word-embeddings (Cimino et al, 2018). These word-embeddings were the most accurate representation of the Italian language at the beginning of our experiment (2019) and won the Best System Award at the EVALITA 2018 (<https://www.evalita.it/evalita-2018/>), which is a periodic evaluation campaign of Natural Language Processing tools for the Italian language.

Second, considering the low-temporal resolution of fMRI (TR of 2 secs), during model definition, we discarded a large set of stop-words (e.g., function words including determiners, conjunctions, prepositions, pronouns, auxiliary and modal verbs), sparing only nouns, verbs, adjectives, adverbs and onomatopoeic words. This approach reduced at least by half the number of averaged word-embeddings per TR, pruning the redundancy of grammatical cues (Mahowald et al., 2022).

That being said, we followed the Reviewer's advice and tested the ability of a more advanced model of language to explain the inter-subject synchronization across sensory modalities. Indeed, as demonstrated in Schrimpf et al., 2021, Word2Vec retained a tenth of the magnitude of brain-model correlations when compared to modern algorithms, as OpenAI GPT. Although in Schrimpf et al., 2021, sentences were fed in word-by-word to generate word-embeddings, we agreed with the Reviewer that the increment of performance of modern transformers accounting for compositionality is unquestionable. Therefore, we performed our analysis again using the most updated version of GPT (i.e., GPT-3).

Considering the availability of the whole verbal content of the movie in English, the narrative was split according to the subtitles, which were mostly based on single complete sentences. Subsequently, we obtained contextual word-embeddings from each sentence using pre-trained GPT-3 (Brown et al., 2020) with OpenAI API (<https://openai.com/api/>) for academic access. GPT-3 is the most advanced artificial language model available nowadays. Its previous version, GPT-2 (Radford et al., 2019), was considered the best biologically feasible framework to predict neural activity during language tasks (Schrimpf et al., 2021; Goldstein et al., 2022). GPT-3 shared the same model and architecture of GPT-2, but consisted of 175 billion parameters, a 100x increase as compared to the previous version.

Briefly, we obtained a vector of 12,288 dimensions from each sentence using the model text-similarity-davinci-001 (available here: <https://beta.openai.com/docs/guides/embeddings>). Sentence-based vectors were stacked in time and resampled to a temporal resolution of 2 seconds, and a PCA was performed to retain at least 90% of the total variance, thus reducing the model to 232 dimensions. Finally, the remaining columns were convolved with a standard gamma function.

We evaluated through voxel-wise encoding the goodness-of-fit of our GPT descriptors in the multisensory AV dataset, and we compared it with Word2Vec performance. Results were described in Supplementary S4, and revealed three important points: 1) the peak of R^2 for GPT features was located in left STG with a magnitude of ~ 0.2 (corresponding to a Pearson's r of ~ 0.4), slightly above the effect of GPT reported by Schrimpf et al., 2021 (please refer to Reviewer 2, point 2 for a detailed comparison); 2) as compared to GPT, Word2Vec retained a similar spatial distribution of the goodness-of-fit, yet with a smaller magnitude; 3) GPT outperforms Word2Vec, with an increment in explained variance up to 7.6%. Overall, these findings demonstrate the quality of the GPT descriptors, although the gain of this modern algorithm was less than the ten-fold increase shown by Schrimpf et al., 2021.

Considering the overall quality of GPT, we defined a new high-level model based on the categorical information and GPT-3 embeddings of English sentences and updated the model-mediated analyses accordingly. Results are reported in Fig.4 of the revised manuscript and are comparable with those based on Word2Vec, showing a consistent engagement of the same portions of STS/STG with also comparable effect sizes in the representation of semantic information. The old high-level model based on Italian word-embeddings obtained through Word2Vec is still available in Supplementary Fig.5.

We now added an entire new paragraph (Computational Models section of Supplementary Materials) to provide all details about the GPT model.

Issue 4. The authors claim that “the overall TRW profile was consistent across the three experimental procedures, indicating that synchronized regions retained a similar chronotopic organization”. This conclusion should be supported by a quantitative analysis, e.g. by correlating the TRW maps across the three procedures.

We thank the Reviewer for this methodological suggestion. The similarity of the TRW maps was evaluated by correlating the TRW maps across the three experimental conditions.

Since the three TRW maps (i.e., AV and A vs V in TD participants, and A vs V in SD participants) had different extensions and encompassed different cortical regions, here our analysis was restricted to the patch of cortex including the voxels showing significant ISC between A vs V conditions in SD participants. This is because all voxels of this map were also included in the other two TRW maps (see Fig.5). In brief, we first estimated Spearman’s ρ across the maps of interest (e.g., between TD and SD for the A vs V condition). To test the statistical significance of this similarity, 200 null TRW maps were generated by means of surrogate signals, resulting in maps with the same spatial smoothness (~6 mm) of the original data. P-values were estimated by comparing the actual correlation coefficient with the ones obtained from the 200 null maps ($p < 0.005$).

Results demonstrated that temporal tunings were coherently represented across all the three experimental conditions, showing high correlations between maps (between A vs V in TD and SD: $\rho = 0.334$; between A vs V in TD and AV in TD: $\rho = 0.431$; between A vs V in SD and AV in TD: $\rho = 0.402$). All correlations were significant, with a $p < 0.005$. For further details, please also refer to Issue 1 of Reviewer 2.

The requested analysis has been added in the revised “Results” and “Methods” sections of the main text.

Issue 5. “The present study questioned whether brain regions involved in audiovisual processing, including the superior temporal and neighboring areas, retain the ability to represent sensory correspondences across modalities, despite the complete lack of any prior auditory or visual experience.” I believe the authors mean “The present study asked...” or “The present study tested...”. The expression “to question something” means to feel or express doubt about that thing, or to raise objections about it.

We thank the Reviewer for this recommendation. We have corrected the above sentence as suggested at page 11 of the revised manuscript:

“The present study tested whether brain regions involved in audiovisual processing, including the superior temporal and neighboring areas, retain the ability to represent sensory

correspondences across modalities, despite the complete lack of prior auditory or visual experience.”

6. In the discussion: “Synchronized neural responses, selectivity for perceptual features and a preserved mapping of temporal receptive windows demonstrate that the functional architecture of the superior temporal cortex emerges despite the lack of audiovisual inputs since birth and irrespectively of postnatal sensory experiences.”

I believe that this claim is too strong and not supported by the results. The way this sentence is phrased, it seems that the authors are saying that all aspects of the functional architecture of superior temporal cortex are independent of experience. However, this cannot be tested with existing methods, because the resolution of functional MRI is not adequate to measure all aspects of the architecture of superior temporal cortex. In addition, as already noted in point 3, the results also reveal substantial differences between the responses in TD and SD participants. The findings are already very interesting without the need of making claims that are not supported by the data. I suggest that the authors moderate this claim and discuss the fact that limits of the instruments mean that many aspects of the functional architecture might not be observable with the approach used.

We have rephrased this sentence, moderating the claim we made in the original version of the manuscript (Discussion section, page 11) as follows:

“Results indicate that a functional architecture of the superior temporal cortex, based on the extraction of common basic features from auditory and visual signals, emerges despite the lack of audiovisual inputs since birth and thus irrespectively of postnatal audiovisual sensory experiences. This observation favors the hypothesis that the human superior temporal cortex is endowed with a functional scaffolding to process low-level perceptual features that define sensory correspondences across audition and vision”.

Issue 7. In the discussion, this sentence “Conversely, the refinement of more complex levels of audiovisual skills appears to require a full multisensory experience throughout lifetime.” does not seem to be directly supported by the results in the article, therefore it needs to be supported by references to the literature. Alternatively, it should be phrased as a hypothesis or speculation rather than as a statement of fact.

This sentence refers to the results of the “model-mediated ISC” and meant to highlight the different impact of low-level and high-level features in driving the synchronization within STS/STG and nearby cortex. Hence, with the expression “more complex levels of audiovisual skills” we specifically referred to the finding that high-level information (i.e., semantic and

categorical movie features) have a role on the synchronization across modalities. However, as correctly suggested by the Reviewer and in light of the discussion of Issue 2, we acknowledge that this sentence has not been directly tested by our experimental protocol and was rephrased and moved to a different section of the Discussion (page 14):

“At the same time, our results also favor the hypothesis that audiovisual experience is required for a full refinement of multisensory functions within temporal and parietal regions. Specifically, in the STS/STG both the reduced synchronization and the reduced representation of higher-level features in SD individuals suggest that sensory experience is necessary for a complete development of their functional specializations. Consistently, even if our study cannot directly evaluate this refinement, audiovisual experience appears to be pivotal for the full development of higher-level computations (e.g., speech perception or semantic congruence)”.

8. “Here, we observed that the high-level movie properties, such as word embeddings and categorical semantic information, mediate the synchronization across auditory and visual stimulations in TD but not in the SD participants. While perceptual features are intrinsic properties of a given stimulus, higher order characteristics rely more on experience and learning, and this may partially account for the observed differences between TD and SD participants.”

This is a surprising result, as categorical semantic information should be the least dependent on changes in perceptual modality. Indeed, previous work (Bedny et al. 2019, see reference below) shows that sensory deprived participants still retain largely intact semantic knowledge – this might well depend on experience and learning, but it appears that even SD participants would have access to the necessary experience (e.g. auditory experience alone might be sufficient to acquire the semantic knowledge). Please reference the relevant previous work and discuss how you might reconcile your results with the previous findings.

Bedny, M., Koster-Hale, J., Elli, G., Yazzolino, L., & Saxe, R. (2019). There’s more to “sparkle” than meets the eye: Knowledge of vision and light verbs among congenitally blind and sighted individuals. Cognition, 189, 105-115.

As correctly observed by the Reviewer, previous behavioral (e.g., Lenci et al., 2013; Bedny et al., 2019) and functional (e.g., Noppeney et al., 2003, Striem-Amit et al., 2018, Mattioni et al., 2020) observations, including studies from our group (e.g., Handjaras et al., 2016, 2017), have revealed similarities (but also some peculiarities) in knowledge organization and semantic representation between congenitally blind and sighted people. Moreover, similarities and as well as differences of semantic representations have been described between congenitally deaf and hearing people (e.g., Capek et al., 2009; Stroh et al., 2019; Cardin et al., 2013).

In these previous works, while the behavioral and functional similarities have been often interpreted as a demonstration of a shared, modality-independent representation of semantic information in the human brain, the differences between the sensory deprived and control groups have been explained as a result of postnatal, experience-dependent reorganization. In addition, all these studies have been classically focused on limited semantic domains and, more importantly, on a distinct sensory deprived group (i.e., “within” a single sensory modality, e.g., audition in the blind people). In our paper, correlations of brain activity were purposely measured across the presentations of visual and auditory narratives, and any evidence of synchronization across conditions and experimental groups has been interpreted as indicative of a shared representations of sensory features despite different postnatal experiences.

On the basis of these observations, the finding that synchronization of the superior temporal cortex in SD is mediated by low-level perceptual features, and not by higher order characteristics, raises two considerations.

First, while we specifically defined a conjunction map aimed at highlighting the regions typically synchronized during audio-visual processing in TD, the activity of the regions that were not synchronized in response to either the audio-only or visual-only conditions, such as early sensory areas, most of the extrastriate regions and large portions of frontal cortex, were not included in the analysis. Semantic processing relies indeed on a large and distributed cortical network. In this regard, even if not reported here because beyond the aim of the current manuscript, semantic knowledge - that is linguistic and categorical information - is still largely preserved (i.e., it has a role in driving the synchronization) within-group, both in blind and deaf participants. For example, a representation of semantic domains is retained in extrastriate regions, such as the lateral occipital complex, the ventral and dorsal stream in congenitally blind individuals, while in early, associative and multimodal auditory regions for the congenitally deaf sample (Setti et al., 2020). Therefore, a wide cortical network expanding beyond the superior temporal cortex does contribute to forming the semantic representation of complex, naturalistic stimuli, despite the lack of information from one sensory modality since birth.

In addition, regarding the superior temporal cortex, it is plausible that distinct post-natal sensory experiences and learning trajectories may trigger different semantic representations of events, therefore hindering the synchronization across blind and deaf individuals, as also explained in Response 1. Additionally, idiosyncratic processes of neural reorganization resulting from a specific form of sensory loss may further hamper the synchronization among sensory deprived individuals. In particular, studies showed that the superior temporal cortex undergoes post-natal functional reorganization in congenitally deaf people during linguistic processing (e.g., Finney et al., 2001; Nishimura et al., 1999; Cardin et al., 2013). On the other hand,

congenital loss of visual input in blindness affects language processing and lateralization of language functions (Lane et al., 2017; Pant et al., 2020).

Therefore, we acknowledge the relevance in the field of additional analyses that will explore the data also from the perspective suggested by the Reviewer. To this aim we would like to say that we made publicly available the fMRI dataset so that the idiosyncratic specific alterations associated with each model of sensory deprivation (i.e., blindness and deafness) could be addressed in future studies.

9. In Figures 2a, 3a, 4 and 5, please include the extreme values for both ends of the color bars (only one extreme is provided).

We have included the extreme values for the color bars in Figs.2-5, as well as in Supplementary Figures, as suggested by the Reviewer.

Reviewer #2:

Remarks to the Author:

In this manuscript, Setti and colleagues present a modified version of the Disney movie "101 Dalmations" (or "Centouno Dalmata" in Italian) and measure the resultant brain activity using fMRI. The movie was presented in either auditory (A), visual (V) or audiovisual (AV) formats. Because the A and V movies are missing information relative to the AV movies, they were supplemented with subtitles for the V movie and with a voice-over describing the missing visual information for the A movie. The movie was presented to separate groups of 10 typically-developed (TD) individuals in A, V, and AV formats; to 10 blind individuals in the A format; and to 10 deaf individuals in the V format (deaf and blind individuals are described as sensory-deprived, SD).

The analysis is based on the idea of intersubject correlation (ISC), measuring whether brain regions that show similar fMRI time courses across participants. ISC can be considered a hybrid between resting-state fMRI analysis methods, which analyze correlations between the fMRI time course of functionally connected regions within a single subject; and task-based fMRI analysis methods, which analyze correlations between an externally-presented stimulus event and the fMRI course of a region.

ISC is well-suited to analyzing the brain response to lengthy or naturalistic stimuli such as movies. The main result from the manuscript is that ISC is found across a good fraction of the brain, especially superior temporal cortex, lateral frontal cortex, and "default mode network" areas. This is true even for movies presented in different

modalities (A vs. V) across TD or SD individuals. The manuscript has many good qualities. The effort necessary to scan both deaf and blind SD individuals clearly sets it apart from most fMRI studies. The ISC analysis is sensible and the finding that temporal cortex synchronizes to movies presented in different modalities is clearly interesting.

We thank the Reviewer for pointing out the positive qualities of our experimental and analytical approaches, including the effort of recruiting and scanning congenitally deprived deaf and blind subjects. We also truly appreciate all the suggestions that contributed to improve the quality of our work. Moreover, we will make our fMRI dataset publicly available as a resource for the whole research community.

Issue 1. The manuscript also has some flaws. The big claim about superior temporal cortex (STC) containing "a modality-independent topographical organization of temporal dynamics" seems only weakly supported by the data. One reason is that fMRI is not an ideal tool for measuring temporal windows. For instance, Luo and Poeppel (2012) use MEG data as evidence that STC contains two temporal windows, one of 20-80 ms, the other 150-300 ms. Given that these window durations are about 100-times faster than the duration of the hemodynamic response measured with BOLD (~15 seconds), it is unlikely that BOLD fMRI can map (or even distinguish) their topographical organization. This is not to say that slow signals in fMRI do not carry information, just that it is a fraught endeavor to tie them a specific neurophysiological property such as the duration of temporal integration windows in STC. As shown in Figure 5, most voxels have a window of ~15 seconds, exactly as you would expect if the window was determined by the autocorrelation induced by the slow BOLD temporal window, rather than any properties of the underlying neurons. In any event, to show that there is a topographical organization, it would be necessary to show a regular organization of window durations within a small region of cortex (akin to a tonotopic map) and the authors do not show any such data. Instead, Figure 5 shows that some areas are measured to have shorter windows, and others are measured to have longer, but this is not equivalent to a topographical organization.

The Reviewer raised two intertwined points, one concerning the low-temporal resolution of BOLD signal, and, more in general, the ability of fMRI to represent fast pacing information; the second related to the organization of such information onto the cortical mantle.

Low-temporal resolution of BOLD signal. Although fMRI certainly does not have comparable temporal resolution to electrophysiological methods (e.g., ECoG, EEG or MEG), several studies show how BOLD signal could still be exploited to get information about the temporal properties of a set of stimuli.

An example of this possibility was provided by Nishimoto et al. (2011). The authors used a model of visual motion (N.B., the very same model we adopted in our paper) to track energy changes of a set of static and dynamic Gabor filters at 2 and 4 Hz during a naturalistic stimulation with a fMRI temporal resolution of 1Hz (TR = 1,000 ms). As known, the BOLD-related physiology reduces the actual resolution to ~0.08 Hz and, theoretically, only signals with a frequency less than 0.04 Hz (at least according to the Nyquist criterion) could be properly reconstructed. Nonetheless, Nishimoto and colleagues were able to predict BOLD responses of stimuli changing up to 4 Hz in the visual cortex, and even to reconstruct the frames of the original movies (15 Hz). This ‘fMRI hyperacuity’ (Op de Beeck, et al., 2010) for the visual motion component critically depends on the varying degree of sparsity of the encoded spatio-temporal information and on its distribution across the cortex (Reddy et al., 2006), with multiple and spatially distant voxels biased towards specific descriptors (Kriegeskorte et al., 2010). These considerations hold for the fMRI decoding of spatial frequencies of the visual system, but recent evidence suggests that the same principles may be also valid for the acoustic domain (e.g., speech and sound processing; Huth et al., 2016, de Heer et al., 2017, Khosla et al., 2021).

Nevertheless, as correctly pointed out by the Reviewer, the temporal constraints of fMRI did not allow a direct match between our coherence plot (Fig.1c), in which the collinearities between the two sensory streams in the movie were evaluated, and the temporal tunings extracted from fMRI brain activity. BOLD response could be considered a ‘temporal filter’ (Kriegeskorte et al., 2010): all stimuli able to evoke an hemodynamic response with a duration of less than a fMRI TR will generate the same temporal tuning (around 15 seconds), thus inflating the number of TRWs at ~15s and, at the same time, hampering the possibility to discriminate among correspondences lasting from tenths to a few seconds. Nonetheless, in our paper, beyond the fact that the most frequent temporal window was ~20 second-long (slightly slower but still compatible with a non-specific autocorrelation induced by the canonical BOLD), other peaks of temporal tunings could be appreciated at ~70 seconds and ~160 seconds (see Figure R1 below).

Figure R1. The figure shows TRWs in A vs V conditions in Typically-Developed (TD) and Sensory-Deprived (SD) participants. Temporal tunings were extracted from the voxels of the TRW map in the A vs V condition in SD individuals (see Fig.5c). To avoid crowding, histogram of TRWs in the AV condition was not shown, although it retained a comparable distribution to the two reported here.

It is likely that long-lasting events have been captured by our ISC analysis across modalities. As shown in our coherence plot (Fig.1c), the movie is also characterized by correspondences across modalities lasting from tens of seconds up to a couple of minutes, when coarse visual properties matched long-lasting auditory features (e.g., ominous music in the dark Cruella De Vil's Manor).

To further corroborate this aspect and as also suggested by Reviewer 1 in Response 4, the consistency of the TRW maps was measured across the three experimental conditions (i.e., multisensory AV, and A vs V in TD and SD individuals). The high correlations between maps demonstrated that temporal tunings were coherently represented across experimental conditions (between A vs V in TD and SD: $\rho = 0.334$; between A vs V in TD and AV in TD: $\rho =$

0.431; between A vs V in SD and AV in TD: $\rho = 0.402$). All correlations were significant, with a non-parametric $p < 0.005$.

Altogether, evidence from Fig.5 and Figure R1 supported the hypothesis that these temporal tunings result from a coupling to the sensory correspondences across modalities, rather than representing a canonical BOLD response detached from the current stimulation. In addition, the consistency of the TRW maps further indicates that the results from the TRW analysis were likely related, although indirectly, to the temporal properties of the stimulus.

Despite the temporal constraints of BOLD response, we still consider that the analytical approaches that characterize the temporal structure of information processing, such as the TRW analysis, may be a useful and additional piece of evidence that demonstrates how different cortical voxels may show multiple temporal tunings. This temporal constraint is anyhow shared with all previous fMRI studies that adopted a similar approach for their main analysis (e.g., Hasson et al., 2008, 2015; Lerner et al., 2011; Baldassano et al., 2017; Blank et al., 2020).

Even more importantly, in our paper, we primarily searched for any evidence of activity synchronization within the superior temporal cortex both across conditions and experimental groups. Then, a thorough description of the synchronized events was provided with a model-mediated version of ISC. Finally, an additional TRW approach characterized the temporal dynamics of the synchronization and offered a complementary description whose results consistently integrate the functional characterization of synchronized brain responses.

To conclude, while we agree with the Reviewer that the temporal information in our study is not comparable to a time-domain analysis performed with ECoG, EEG or MEG, the evidence provided above supports the feasibility of this technique in fMRI. Nevertheless, on these issues, a new paragraph has been added to the Limitations section in Supplementary Materials (page 7), to acknowledge the temporal reliability of these estimates.

The organization of TRW information onto the cortical surface. The term “topographical organization” is meant to indicate a hierarchy of TRW evoked by the processing of naturalistic sensory information. This term has been widely used in the fMRI field (e.g., Hasson et al., 2008, 2015; Lerner et al., 2011; Baldassano et al., 2017; Blank et al., 2020) and refers to the idea that different brain regions process sensory information accumulating over time with a specific temporal tuning. However, when compared to classical feature-based maps in primary sensory cortex and other high-level localized regions (Lettieri et al., 2019; Protopapa et al., 2019), our topographical organization, as well as the other maps cited in the literature above, are coarser and encompass multiple brain regions. This large-scale organization limits the possibility to properly test the existence of a canonical gradient-like topography, since multiple and distant

clusters with the same temporal tuning cancel out the overall association between functional and anatomical distance (Yarrow et al., 2014). However, we believe that, although loosely-defined, the current notion of topography, where anatomically adjacent voxels enclose neurons with similar functional properties, is still appropriate for our maps in Fig.5. The current definition of “topographical organization” has been added in the Discussion section (page 13) of the revised manuscript to avoid any confusion with the localized feature-based maps.

2. One disadvantage of ISC is that it can be difficult to know what exactly in the movie is triggering the observed correlation between subjects. The properties of the movie can be examined in an attempt to explain observed correlations. For instance, in the original Hasson et al. paper, movie segments containing faces evoked strong ISC in the fusiform face area (presumably since all subjects' FFAs showed strong responses at these times). Setti and colleagues use a quantitative modeling approach to measure the effect of low-level visual features, low-level acoustic features, and high-level semantic features, calculated from the stimulus movie. In general, this approach is sensible and sophisticated, modeling low-level sensory features and high-level semantic features. However, it is not exactly clear how well the models worked. Figure S1 shows some data on how well the models predicted the observed MR time series. In the referenced study of Nishimoto et al., the correlation between predicted and observed BOLD fMRI was as high as $r = 0.40$. In the current study, it seems that the values are quite a bit lower. It would be good to compare the model fit values between the current study and previous studies that used similar methods (many studies from Gallant and colleagues). It is also difficult to interpret the "model-mediated ISC", that is, the drop in ISC, which seems quite low (1-2%). It is not good to draw firm conclusions based on this small change and dichotomize areas as being responsive (or not) to the features in a model based on a tiny drop (or not) in ISC correlation, due to the statistical fallacy that the difference between "significant" (e.g. $p = 0.04$) and "not significant" (e.g. $p = 0.06$) is often not itself significant. The preferred method would be to perform a direct statistical comparison between the goodness-of-fit (between the modeled and measured fMRI time series) for models with and without a given feature included (the method used by Gallant and colleagues to compare different models).

We thank the Reviewer for highlighting this crucial aspect. Here, we provide a detailed response to this issue. First, we describe the alternative methods available to understand the role of a set of computational descriptors in explaining brain activity (i.e., Hasson's reverse-correlation, Gallant's encoding, mediation analysis). Since our approach both relied on mediation (for the A vs V conditions) and encoding (for the AV condition), the goodness-of-fit of our models was tested with encoding and their performance was compared with the ones

reported in the current literature. Then, the major issue of the low-effect size was addressed by demonstrating its dependence on collinearities between models which were removed to facilitate findings interpretability. Finally, the difference between low- and high-level models was tested in TD and SD participants to further corroborate the relatively high contribution of visual and acoustic features in driving the ISC between sensory modalities.

Alternative methods to understand the role of computational descriptors in explaining brain activity. In their seminal work, Hasson et al. (2004) used a procedure called ‘reverse correlation’ to understand the functional role of different brain regions. Considering also other successful applications (e.g., Richardson et al., 2018), we preliminarily used such an approach to infer the features associated with the correspondences across sensory modalities. This methodology crucially depends on a few factors:

- the effect size of ISC must be high (i.e., $r \sim 0.4$ or above), as in our AV multisensory condition and in other ISC studies with naturalistic stimulation (e.g., Oleggio Castello et al., 2020), whereas correlation coefficients in the A vs V experimental condition reaches half of this magnitude;
- the method is successful only when a solid evidence of the function of a brain region exists (e.g., FFA for Faces, TPJ for emotions), which may not be the case for our regions of interest (e.g., STS/STG, PCC);
- ‘reverse correlation’ is substantially based on subjective evaluations and this aspect may be critical when dealing with a theoretical set of low-level descriptors (e.g., motion energy at specific spatial and temporal frequencies, acoustic spectrogram).

Thus, for the above reasons, we limited such analysis to the AV condition and for an internal evaluation only to verify a higher synchronization in PPA and FFA during place and face processing, respectively (results not shown here).

Then we took into consideration the ‘encoding approach’ used by Gallant and colleagues (Naselaris et al. 2011). The encoding procedure is designed to predict brain activity within the single subject, by modeling the maximum fraction of explainable variance. Therefore, this procedure could completely rely on idiosyncratic neural responses and it is not suitable for measuring the shared activity across individuals and experimental conditions, which was the goal of our study. To overcome this limitation, in our study we used a modified version of Gallant's encoding for the AV experimental condition. Specifically, we applied a group-level encoding procedure obtaining a measure of predictability (i.e., cross-validated leave-one-run-out R^2) and identified the best feature sets across TD participants in the AV condition. These optimal sets of descriptors, with equal dimensionality and high predictive power, were then used in the independent A vs V datasets to test our hypothesis.

Here, we report the goodness-of-fit of our models as Figure R2a below and as Supplementary Fig.1a in the manuscript.

Figure R2. Figure depicts the results of the encoding procedure on the AV condition, for the low-level visual, low-level acoustic, high-level Word2Vec semantic and the high-level GPT-3 semantic models, respectively. Panel **a** represents the explained variance for the original models, whereas Panel **b** shows the performance of models when a large portion of common variance was removed (i.e., movie editing

features). Unthresholded cross-validated R^2 maps (cross-validated $R^2 > 0$, minimum cluster size of 20 adjacent voxels).

Specifically, low-level visual and acoustic features, the high-level semantic descriptor and the newest high-level GPT-3 semantic model (please refer to Issue 3 of Reviewer 1 for further details) were tested. As far as the goodness-of-fit is concerned, the peaks of cross-validated R^2 (averaged across subjects) were reported for low-level visual features (right Superior Occipital Gyrus, $R^2=0.127$), low-level auditory features (left STG, $R^2=0.290$), high-level semantic (categorical and Word2Vec) features (left STG, $R^2=0.195$), high-level semantic (categorical and GPT-3) features (left STG, $R^2=0.231$).

The effect sizes of the computational models in the AV condition were aligned with the current fMRI literature. As suggested by the Reviewer, we compared the goodness-of-fit of the low-level visual model in our study with the one reported by Nishimoto et al. (2011). Although Nishimoto et al.'s and our experiments had different setups (i.e., 4T vs 3T, partial vs total brain coverage, TR 1s vs 2s, voxel size of 2x2x2.5 vs 3x3x3 mm, short movies up to 20s vs one long movie, 10 repetitions for each short movie vs a movie shown only once, respectively), a coarse comparison between the two experiments is provided here.

In Nishimoto et al., the motion energy model performed $r = 0.40$ in a selection of active voxels in early visual areas (V1, V2, V3, V3A, and V3B), as defined through a localizer and thresholded with an unspecified procedure to select the best performing 2,000 voxels. Considering that we shared a similar visual angle of the stimulation (20° vs 22°), so that early visual activity should be distributed onto the same cortical space, we identified the same volume of top performing voxels in early visual areas (defined through the visfAtlas, Rosenke et al., 2021). Within this region, we obtained an average $r = 0.202$ (min $r=0.150$, max= 0.356). Since Nishimoto et al. had a higher SNR due to the 10 repetitions for each movie, we performed our encoding procedure to predict the left-run-out signal averaged across 10 subjects. Although the 10x gain of Nishimoto et al. was reached by averaging repetitions to reduce within-subject variability, while here brain activity was averaged to reduce between-subject variability, we considered this procedure a more fair comparison with the reported effect size. Within the same region, an average $r=0.349$ (min $r=0.222$, max= 0.540) was obtained, in line with the Nishimoto et al.'s $r=0.40$.

In addition to the visual model, we also assessed the goodness-of-fit of the low-level acoustic model. Indeed, this set of descriptors included a few columns related to the sound envelope properties (Martinelli et al., 2020) and 449-dimensions shared with a power-spectrum descriptor defined in de Heer et al. (2017). In this case, de Heer et al.'s and our experiment had

a more similar setup as compared to the previous comparison: both 3T and TR 2s, voxel size of 2.24x2.24x4.1 vs 3x3x3 mm, only auditory vs audiovisual stimulation, different stories up to 15 minutes vs one long movie, both no repetitions. In de Heer et al., the model performed $r=0.244$ in a selection of active voxels in the left early acoustic cortex, defined through a localizer, and thresholded with an unspecified procedure. Nonetheless, we inspected Fig.3 reported in de Heer et al.'s paper and defined a ROI of similar position and extension using the HCP atlas (Glasser et al., 2016). Within this region, in de Heer's paper voxels were further selected according to $p<0.01$, FDR-corrected. Here, we selected only voxels with a cross-validated $R^2>0$ and then estimated the correlation coefficient across voxels, which resulted to be on average $r=0.280$ (min $r=0.01$, max=0.538), in line with the originally reported $r=0.244$.

Finally, we tested the goodness-of-fit of the high-level GPT-3 semantic model (based on categorical information and GPT-3 embeddings of English sentences) with the results reported in Schrimpf et al. (2021), which assessed the performance of state-of-the-art artificial-neural-network language models in different fMRI and ECoG datasets. Specifically, following the 'Materials and Methods' of Schrimpf et al. (2021), within the language network defined in Fedorenko et al., (2010), the peak activity at voxel level was compared with the *Pereira 2018* dataset ($r=0.32$) and the average correlation at ROI-level with the *Blank 2014* dataset ($r=0.20$). Further details of these datasets can be retrieved in the original manuscript of Schrimpf et al. (2021). The ceiling effects reported by Schrimpf and colleagues were related to the GPT-2 set of features, whereas here our model consisted of a few columns from the categorical descriptors and hundreds from GPT-3 (Brown et al., 2020). GPT-3 from OpenAI (<https://openai.com/>) is the most advanced artificial language model available in 2022 (see Issue 3 of Reviewer 1). Its previous version, GPT-2 (Radford et al., 2019) was considered the best biologically feasible framework to predict neural activity during language tasks (Schrimpf et al., 2021; Goldstein et al., 2022). GPT-3 shared the same model and architecture of GPT-2, but consisted of 175 billion parameters, a 100x increase as compared to the previous version.

When compared to the *Pereira 2018* dataset ($r=0.32$), our peak performance at voxel level was $r=0.424$, whereas the average correlation in the language network was $r=0.178$, in line with the effect found by Schrimpf and colleagues using the *Blank2014* dataset ($r=0.20$).

Altogether, our computational set of descriptors are in line with current literature.

Collinearities between our computational models and movie editing features. Before testing whether and to what extent shared neural activity in the A vs V conditions was ascribable to one of these sets of descriptors, we investigated model collinearities. First, we identified a set of features (i.e., seven binary columns) resulting from the editing procedure: the onset of visual cuts, the scene transitions, the presence of subtitles, text embedded in the visual frames, audio

descriptions, dialogues and soundtracks. As reported in Supplementary Fig.2b, this set of coarse descriptors shared a large portion of variance with all the other models.

We also tested the goodness-of-fit of this set of movie editing features in the fMRI AV condition, obtaining a peak of R^2 within left STG of 0.285 ($r \sim 0.534$; see Supplementary Fig.2a). This result demonstrates that a very large portion of fMRI activity represents the movie editing descriptors. In addition, these descriptors had a low computational complexity and a limited capacity to provide a detailed description of the neural systems and cognitive processes involved in stimulus processing. This aspect has been extensively discussed in the Supplementary Materials of the revised manuscript (pages 7-8).

Since these movie editing descriptors were captured by any other high- or low-level model, to facilitate interpretability, each model was orthogonalized with respect to this set of features. After this step, the overall quality of these “cleaned” models in an encoding procedure was re-tested to predict brain activity of fMRI data of the multisensory AV condition (panel b in Figure R2 above and in Supplementary Fig.1b). The removal of such features determined a consistent drop in the peaks of cross-validated R^2 , from 0.127 to 0.117 for the low-level visual model, 0.290 to 0.043 for the low-level acoustic model, 0.231 to 0.080 for the high-level semantic model. On the other hand, the spatial specificity of the cortical mapping of these computational features was improved. Specifically:

- A. the fitting of the visual features was restricted to the early visual cortex;
- B. the acoustic descriptors were now localized to the early auditory cortex, STS/STG, and with a limited engagement of visual cortices likely due to audiovisual low-level collinearities (which were at the basis of our hypothesis);
- C. the high semantic model was specifically represented in high-order visual (LOC, IT), parieto-occipital (POS), and temporo-occipital (pSTG/pSTS, pMTG) regions, in line with recent evidence on the organization of multimodal semantic knowledge (Popham et al., 2021);
- D. As described in Supplementary Fig.2a, the movie editing features were able to explain brain activity across the whole cortex, ranging from the visual and auditory regions, as well as ventral and dorsal attention networks in the frontal and parietal lobes (Anderson et al., 2006), and the language network (Fedorenko et al., 2010). This piece of evidence suggested that movie editing comprised features with a low computational specificity, making this set of covariates an ideal candidate to clean up the low- and high-level models.

The low effect sizes in the model-mediated ISC resulted from the removal of movie editing features. In light of this, the “cleaned” models were then applied to our model-mediated ISC analysis. We specifically decided to adopt the framework of mediation analysis instead of the classical encoding approach, because our analysis focused on the common variance across the A and V conditions, and ISC is the optimal tool to highlight these similarities. Moreover, since our measure of effect size was the ISC coefficient, the drop of ISC (i.e., the mediation effect of a computation model) directly relates to our main measure of effect size.

Nevertheless, we agreed with the Reviewer that the drop in model-mediated ISC is quite small as compared to the ceiling of the effect (i.e., the “classical” ISC). Therefore, we first assessed the mediation effect of the movie-editing features alone and subsequently measured the impact of the cleaning procedure on our computational models.

Figure R3. In Panel **a**, we report the model-mediated ISC across TD in the A vs V condition for the set of movie editing features. In Panel **b**, the same mediation effect is represented across SD participants. All results were corrected for multiple comparisons ($p < 0.05$, Family-wise Error corrected -FWEc-, minimum cluster size of 20 adjacent voxels).

As a matter of fact, the movie editing descriptor accounted for about half of the intersubject correlation (see Figure R3 and Supplementary Fig.3). For instance, in A vs V in TD participants, “classical” ISC at the peak in left STS passed from $r \sim 0.214$ to a model-mediated ISC of $r \sim 0.127$ with an effect of the movie editing features of $r \sim 0.087$. Moreover, the magnitude of the movie

editing features was on average four-fold as compared to the other models. Considering the same condition, in A vs V in TD participants, the mediation effect for the low-level model in left STS was only $r \sim 0.018$.

We also tested the differences of the effect size of the computational models before and after the removal of the movie editing features. Indeed, model-mediated ISC accounted on average for one third of the original magnitude: in A vs V in TD participants, mediation for the low-level model in left STS passed from $r \sim 0.105$ to $r \sim 0.017$ when moving from the original descriptors to the cleaned ones. Although the small effect size of the cleaned models had a significant impact in mediation of ISC, one could consider this effect as negligible. Conversely, we would like to emphasize that:

- A. the orthogonalization procedure was intended to favor the interpretability of the mediation effect;
- B. the set of coarse features included in the movie editing should not be considered as “computational modeling”, since they retained low complexity and limited capacity to describe neural systems and cognitive processes;
- C. movie editing descriptors represented a coarse set of collinearities across senses, and their relatively large effect size in STS/STG further corroborated the hypothesis that these regions still responded to correspondences across modalities.

Moreover, we would like to highlight these additional points:

- D. this issue is intrinsic to the naturalistic stimulation itself, since such a paradigm captures information that is often shared across distinct sensory modalities (please refer to Grall et al., 2022 for a detailed discussion on the role of formal cinematic features as the editing technique);
- E. during the model testing and feature selection procedures in the AV multisensory condition (Supplementary Fig.1), the cleaned models were still able to explain a reasonable portion of variance and the cortical distribution of all sets of features is consistent with previous observations, even for the multimodal high-level representations (Popham et al., 2021);
- F. beyond early visual areas, the ability to explain brain activity through computational models is far from reaching the noise ceiling (Khaligh-Razavi et al., 2014).

Model-mediated differences in ISC between low- and high-level models. Finally, to corroborate our findings on the relative specificity of the low-level models as compared to the high-level semantic descriptors, we tested whether the model-mediated ISC differed in our regions of

interest (see the revised version of Figs.4e-f in the current version of the manuscript). Results in both TD and SD participants indicated that the mediation of low-level features had a higher magnitude ($p < 0.05$, FWEc) in posterior and middle STS/STG, whereas, in their anterior portions, this effect was limited to TD individuals only. On the other hand, in both TD and SD groups, the high-level features exhibited a significantly higher effect ($p < 0.05$, FWEc) in a small patch of cortex, centered around bilateral TPOJ, PG and POS regions.

To conclude, we provided evidence that:

- A. the effect sizes of the computational models in the AV condition were aligned with the current fMRI literature;
- B. the low effect sizes in the model-mediated ISC were not dependent on the specific algorithm, but instead were the results of our cleaning procedure to remove movie editing features;
- C. the removal of movie editing descriptors improved the interpretability of the relative contribution of low- and high-level features in the synchronization procedure.

We added an entire new paragraph in the Limitations (Supplementary Materials), and in Results, Discussion and Methods in the main manuscript to address these points, as well as a new Supplementary Fig.1, Supplementary Fig.3, and Fig.4.

3. Minor notes

line 757: "outern"?

We thank the Reviewer for having highlighted this typo. The term has been replaced by "outer", which we used to indicate a loop that has another loop nested inside it.

References

- Anderson, D. R., Fite, K. V., Petrovich, N., & Hirsch, J. (2006). Cortical activation while watching video montage: An fMRI study. *Media Psychology*, 8(1), 7-24.
- Baldassano, C., Chen, J., Zadbood, A., Pillow, J. W., Hasson, U., & Norman, K. A. (2017). Discovering event structure in continuous narrative perception and memory. *Neuron*, 95(3), 709-721.
- Bedny, M., Koster-Hale, J., Elli, G., Yazzolino, L., & Saxe, R. (2019). There's more to "sparkle" than meets the eye: Knowledge of vision and light verbs among congenitally blind and sighted individuals. *Cognition*, 189, 105-115.

- Blank, I. A., & Fedorenko, E. (2020). No evidence for differences among language regions in their temporal receptive windows. *NeuroImage*, 219, 116925.
- Brown, T., Mann, B., Ryder, N., Subbiah, M., Kaplan, J. D., Dhariwal, P., ... & Amodei, D. (2020). Language models are few-shot learners. *Advances in neural information processing systems*, 33, 1877-1901.
- Capek, C. M., Grossi, G., Newman, A. J., McBurney, S. L., Corina, D., Roeder, B., & Neville, H. J. (2009). Brain systems mediating semantic and syntactic processing in deaf native signers: Biological invariance and modality specificity. *Proceedings of the National Academy of Sciences*, 106(21), 8784-8789.
- Cardin, V., Orfanidou, E., Rönnerberg, J., Capek, C. M., Rudner, M., & Woll, B. (2013). Dissociating cognitive and sensory neural plasticity in human superior temporal cortex. *Nature communications*, 4(1), 1-5.
- Chandrasekaran, C., Trubanova, A., Stillitano, S., Caplier, A., & Ghazanfar, A. A. (2009). The natural statistics of audiovisual speech. *PLoS computational biology*, 5(7), e1000436.
- Chang, C. H., Nastase, S. A., & Hasson, U. (2021). Information flow across the cortical timescales hierarchy during narrative comprehension. *BioRxiv*.
- Cimino, A., De Mattei, L., & Dell'Orletta, F. (2018). Multi-task learning in deep neural networks at evalita 2018. *Proceedings of the 6th evaluation campaign of Natural Language Processing and Speech tools for Italian (EVALITA'18)*, 86-95.
- de Beeck, H. P. O. (2010). Against hyperacuity in brain reading: spatial smoothing does not hurt multivariate fMRI analyses?. *Neuroimage*, 49(3), 1943-1948.
- de Heer, W. A., Huth, A. G., Griffiths, T. L., Gallant, J. L., & Theunissen, F. E. (2017). The hierarchical cortical organization of human speech processing. *Journal of Neuroscience*, 37(27), 6539-6557.
- Desai, S., Stickney, G., & Zeng, F. G. (2008). Auditory-visual speech perception in normal-hearing and cochlear-implant listeners. *The Journal of the Acoustical Society of America*, 123(1), 428-440.
- Fedorenko, E., Hsieh, P. J., Nieto-Castañón, A., Whitfield-Gabrieli, S., & Kanwisher, N. (2010). New method for fMRI investigations of language: defining ROIs functionally in individual subjects. *Journal of neurophysiology*, 104(2), 1177-1194.
- Finney, E. M., Fine, I., & Dobkins, K. R. (2001). Visual stimuli activate auditory cortex in the deaf. *Nature neuroscience*, 4(12), 1171-1173.

- Gibson, E. J. (1970). The development of perception as an adaptive process: Perception of events in space develops early, while perception of objects shows evolution and greater dependence on learning. *American Scientist*, 58(1), 98-107.
- Glasser, M. F., Coalson, T. S., Robinson, E. C., Hacker, C. D., Harwell, J., Yacoub, E., ... & Van Essen, D. C. (2016). A multi-modal parcellation of human cerebral cortex. *Nature*, 536(7615), 171-178.
- Goldstein, A., Zada, Z., Buchnik, E., Schain, M., Price, A., Aubrey, B., ... & Hasson, U. (2022). Shared computational principles for language processing in humans and deep language models. *Nature neuroscience*, 25(3), 369-380.
- Grall, C., & Finn, E. S. (2022). Leveraging the power of media to drive cognition: a media-informed approach to naturalistic neuroscience. *Social Cognitive and Affective Neuroscience*, 17(6), 598-608.
- Guerreiro, M. J., Putzar, L., & Röder, B. (2015). The effect of early visual deprivation on the neural bases of multisensory processing. *Brain*, 138(6), 1499-1504.
- Handjaras, G., Ricciardi, E., Leo, A., Lenci, A., Cecchetti, L., Cosottini, M., Marotta, G., & Pietrini, P. (2016). How concepts are encoded in the human brain: A modality independent, category-based cortical organization of semantic knowledge. *NeuroImage*, 135.
- Handjaras, G., Leo, A., Cecchetti, L., Papale, P., Lenci, A., Marotta, G., ... & Ricciardi, E. (2017). Modality-independent encoding of individual concepts in the left parietal cortex. *Neuropsychologia*, 105, 39-49.
- Hasson, U., Nir, Y., Levy, I., Fuhrmann, G., & Malach, R. (2004). Intersubject synchronization of cortical activity during natural vision. *science*, 303(5664), 1634-1640.
- Hasson, U., Yang, E., Vallines, I., Heeger, D. J., & Rubin, N. (2008). A hierarchy of temporal receptive windows in human cortex. *Journal of Neuroscience*, 28(10), 2539-2550.
- Hasson, U., Chen, J., & Honey, C. J. (2015). Hierarchical process memory: memory as an integral component of information processing. *Trends in cognitive sciences*, 19(6), 304-313.
- Hillock-Dunn, A., & Wallace, M. T. (2012). Developmental changes in the multisensory temporal binding window persist into adolescence. *Developmental science*, 15(5), 688-696.

- Huth, A. G., De Heer, W. A., Griffiths, T. L., Theunissen, F. E., & Gallant, J. L. (2016). Natural speech reveals the semantic maps that tile human cerebral cortex. *Nature*, 532(7600), 453-458.
- Huyse, A., Berthommier, F., & Leybaert, J. (2013). Degradation of labial information modifies audiovisual speech perception in cochlear-implanted children. *Ear and hearing*, 34(1), 110-121.
- Khaligh-Razavi, S. M., & Kriegeskorte, N. (2014). Deep supervised, but not unsupervised, models may explain IT cortical representation. *PLoS computational biology*, 10(11), e1003915.
- Khosla, M., Ngo, G. H., Jamison, K., Kuceyeski, A., & Sabuncu, M. R. (2021). Cortical response to naturalistic stimuli is largely predictable with deep neural networks. *Science Advances*, 7(22), eabe7547.
- Kriegeskorte, N., Cusack, R., & Bandettini, P. (2010). How does an fMRI voxel sample the neuronal activity pattern: compact-kernel or complex spatiotemporal filter?. *Neuroimage*, 49(3), 1965-1976.
- Kuhl, P. K., & Meltzoff, A. N. (1982). The bimodal perception of speech in infancy. *Science*, 218(4577), 1138-1141.
- Lane, C., Kanjlia, S., Richardson, H., Fulton, A., Omaki, A., & Bedny, M. (2017). Reduced left lateralization of language in congenitally blind individuals. *Journal of cognitive neuroscience*, 29(1), 65-78.
- Lenci, A., Baroni, M., Cazzolli, G., & Marotta, G. (2013). BLIND: A set of semantic feature norms from the congenitally blind. *Behavior research methods*, 45(4), 1218-1233.
- Lerner, Y., Honey, C. J., Silbert, L. J., & Hasson, U. (2011). Topographic mapping of a hierarchy of temporal receptive windows using a narrated story. *Journal of Neuroscience*, 31(8), 2906-2915.
- Lettieri, G., Handjaras, G., Ricciardi, E., Leo, A., Papale, P., Betta, M., ... & Cecchetti, L. (2019). Emotionotopy in the human right temporo-parietal cortex. *Nature communications*, 10(1), 1-13.
- Lewkowicz, D. J. (2000). The development of intersensory temporal perception: an epigenetic systems/limitations view. *Psychological bulletin*, 126(2), 281.
- Lewkowicz, D. J. (2010). Infant perception of audio-visual speech synchrony. *Developmental psychology*, 46(1), 66.

- Lewkowicz, D. J., & Turkewitz, G. (1980). Cross-modal equivalence in early infancy: Auditory-visual intensity matching. *Developmental psychology*, 16(6), 597-607.
- Mahowald, K., Diachek, E., Gibson, E., Fedorenko, E., & Futrell, R. (2022). Grammatical cues are largely, but not completely, redundant with word meanings in natural language. *arXiv preprint arXiv:2201.12911*.
- Martinelli, A., Handjaras, G., Betta, M., Leo, A., Cecchetti, L., Pietrini, P., ... & Bottari, D. (2021). Auditory features modelling reveals sound envelope representation in striate cortex. *bioRxiv*, 2020-04.
- Mattioni, S., Rezk, M., Battal, C., Bottini, R., Mendoza, K. E. C., Oosterhof, N. N., & Collignon, O. (2020). Categorical representation from sound and sight in the ventral occipito-temporal cortex of sighted and blind. *ELife*, 9.
- Munhall, K. G., & Vatikiotis-Bateson, E. (2004). Spatial and temporal constraints on audiovisual speech perception. In *The Handbook of Multisensory Processes*.
- Murray, M. M., Lewkowicz, D. J., Amedi, A., & Wallace, M. T. (2016). Multisensory processes: a balancing act across the lifespan. *Trends in Neurosciences*, 39(8), 567-579.
- Naselaris, T., Kay, K. N., Nishimoto, S., & Gallant, J. L. (2011). Encoding and decoding in fMRI. *Neuroimage*, 56(2), 400-410.
- Nishimoto, S., Vu, A. T., Naselaris, T., Benjamini, Y., Yu, B., & Gallant, J. L. (2011). Reconstructing visual experiences from brain activity evoked by natural movies. *Current Biology*, 21(19).
- Noppeney, U., Friston, K. J., & Price, C. J. (2003). Effects of visual deprivation on the organization of the semantic system. *Brain*, 126(7), 1620-1627.
- Neville, H. J., Bavelier, D., Corina, D., Rauschecker, J., Karni, A., Lalwani, A., Braun, A., Clark, V., Jezzard, P., & Turner, R. (1998). Cerebral organization for language in deaf and hearing subjects: Biological constraints and effects of experience. *Proceedings of the National Academy of Sciences of the United States of America*, 95(3).
- Nishimura, H., Hashikawa, K., Iwaki, T., Watanabe, Y., Kusuoka, H., Nishimura, T., & Kubo, T. (1999). Sign language 'heard' in the auditory cortex. *Nature*, 397(6715), 116-116.
- Pant, R., Kanjlia, S., & Bedny, M. (2020). A sensitive period in the neural phenotype of language in blind individuals. *Developmental Cognitive Neuroscience*, 41.
- Patterson, M. L., & Werker, J. F. (2003). Two-month-old infants match phonetic information in lips and voice. *Developmental Science*, 6(2), 191-196.

- Popham, S. F., Huth, A. G., Bilenko, N. Y., Deniz, F., Gao, J. S., Nunez-Elizalde, A. O., & Gallant, J. L. (2021). Visual and linguistic semantic representations are aligned at the border of human visual cortex. *Nature neuroscience*, *24*(11), 1628-1636.
- Protopapa, F., Hayashi, M. J., Kulashekhar, S., van der Zwaag, W., Battistella, G., Murray, M. M., ... & Bueti, D. (2019). Chronotopic maps in human supplementary motor area. *PLoS Biology*, *17*(3), e3000026.
- Putzar, L., Goerendt, I., Lange, K., Rösler, F., & Röder, B. (2007). Early visual deprivation impairs multisensory interactions in humans. *Nature neuroscience*, *10*(10), 1243-1245.
- Putzar, L., Gondan, M., & Röder, B. (2012). Basic multisensory functions can be acquired after congenital visual pattern deprivation in humans. *Developmental neuropsychology*, *37*(8), 697-711.
- Radford, A., Wu, J., Child, R., Luan, D., Amodei, D., & Sutskever, I. (2019). Language models are unsupervised multitask learners. *OpenAI blog*, *1*(8), 9.
- Reddy, L., & Kanwisher, N. (2006). Coding of visual objects in the ventral stream. *Current opinion in neurobiology*, *16*(4), 408-414.
- Richardson, H., Lisandrelli, G., Riobueno-Naylor, A., & Saxe, R. (2018). Development of the social brain from age three to twelve years. *Nature communications*, *9*(1), 1-12.
- Rosenke, M., van Hoof, R., van den Hurk, J., Grill-Spector, K., & Goebel, R. (2021). A probabilistic functional atlas of human occipito-temporal visual cortex. *Cerebral Cortex*, *31*(1), 603-619.
- Rouger, J., Fraysse, B., Deguine, O., & Barone, P. (2008). McGurk effects in cochlear-implanted deaf subjects. *Brain research*, *1188*, 87-99.
- Schrimpf, M., Blank, I. A., Tuckute, G., Kauf, C., Hosseini, E. A., Kanwisher, N., ... & Fedorenko, E. (2021). The neural architecture of language: Integrative modeling converges on predictive processing. *Proceedings of the National Academy of Sciences*, *118*(45), e2105646118.
- Schorr, E. A., Fox, N. A., van Wassenhove, V., & Knudsen, E. I. (2005). Auditory-visual fusion in speech perception in children with cochlear implants. *Proceedings of the National Academy of Sciences*, *102*(51), 18748-18750.
- Setti F., Handjaras G., Diano M., Bruno V., Tinti C., Pietrini P., Garbarini F., Leo A., Ricciardi E (2020). Naturalistic stimulation in sensory-deprived individuals reveals different reorganization mechanisms. 2020 Organization for Human Brain Mapping Annual Meeting (online).

- Setti, F. (2020). Naturalistic stimulation in sensory-deprived individuals reveals overlapping large-scale brain organization with differential cross-modal mechanisms. [IMT PhD Thesis].
- Stein, B. E., Stanford, T. R., & Rowland, B. A. (2014). Development of multisensory integration from the perspective of the individual neuron. *Nature Reviews Neuroscience*, *15*(8), 520-535.
- Striem-Amit, E., Wang, X., Bi, Y., & Caramazza, A. (2018). Neural representation of visual concepts in people born blind. *Nature communications*, *9*(1), 1-12.
- Stroh, A. L., Rösler, F., Dormal, G., Salden, U., Skotara, N., Hänel-Faulhaber, B., & Röder, B. (2019). Neural correlates of semantic and syntactic processing in German Sign Language. *Neuroimage*, *200*, 231-241.
- Stropahl, M., Plotz, K., Schönfeld, R., Lenarz, T., Sandmann, P., Yovel, G., ... & Debener, S. (2015). Cross-modal reorganization in cochlear implant users: Auditory cortex contributes to visual face processing. *Neuroimage*, *121*, 159-170.
- Tona, R., Naito, Y., Moroto, S., Yamamoto, R., Fujiwara, K., Yamazaki, H., ... & Kikuchi, M. (2015). Audio-visual integration during speech perception in prelingually deafened Japanese children revealed by the McGurk effect. *International journal of pediatric otorhinolaryngology*, *79*(12), 2072-2078.
- Tremblay, C., Champoux, F., Lepore, F., & Théoret, H. (2010). Audiovisual fusion and cochlear implant proficiency. *Restorative neurology and neuroscience*, *28*(2), 283-291.
- Trettenbrein, P. C., Papitto, G., Friederici, A. D., & Zaccarella, E. (2021). Functional neuroanatomy of language without speech: An ALE meta-analysis of sign language. *Human Brain Mapping*, *42*(3).
- Visconti di Oleggio Castello, M., Chauhan, V., Jiahui, G., & Gobbini, M. I. (2020). An fMRI dataset in response to “The Grand Budapest Hotel”, a socially-rich, naturalistic movie. *Scientific Data*, *7*(1), 1-9.
- Walton, G. E., & Bower, T. G. R. (1993). Amodal representation of speech in infants. *Infant Behavior and Development*, *16*(2), 233-243.
- Yarrow, S., Razak, K. A., Seitz, A. R., & Series, P. (2014). Detecting and quantifying topography in neural maps. *PLoS one*, *9*(2), e87178.
- Zhou, H. Y., Cheung, E. F., & Chan, R. C. (2020). Audiovisual temporal integration: Cognitive processing, neural mechanisms, developmental trajectory and potential interventions. *Neuropsychologia*, *140*, 107396.

Decision Letter, first revision:

10th October 2022

Dear Dr. Ricciardi,

Thank you for submitting your revised manuscript "A modality independent proto-organization of human multisensory areas" (NATHUMBEHAV-22030748A). It has now been seen by the original referees and their comments are below. As you can see, the reviewers find that the paper has improved greatly in revision. We will therefore be happy in principle to publish it in Nature Human Behaviour, pending changes to comply with our editorial and formatting guidelines.

We are now performing detailed checks on your paper and will send you a checklist detailing our editorial and formatting requirements within a week. Please do not upload the final materials and make any revisions until you receive this additional information from us.

Sincerely,
Jamie

Dr Jamie Horder
Senior Editor
Nature Human Behaviour

Reviewer #1 (Remarks to the Author):

The authors have addressed my concerns in full.

Reviewer #2 (Remarks to the Author):

The authors have done an excellent job of addressing my comments and I have no additional requests.

Sincerely,
Mike Beauchamp

Final Decision Letter:

Dear Professor Ricciardi,

We are pleased to inform you that your Article "A modality independent proto-organization of human multisensory areas", has now been accepted for publication in *Nature Human Behaviour*.

Please note that *Nature Human Behaviour* is a Transformative Journal (TJ). Authors whose manuscript was submitted on or after January 1st, 2021, may publish their research with us through the traditional subscription access route or make their paper immediately open access through payment of an article-processing charge (APC). Authors will not be required to make a final decision about access to their article until it has been accepted. IMPORTANT NOTE: Articles submitted before January 1st, 2021, are not eligible for Open Access publication. Find out more about Transformative Journals

An online order form for reprints of your paper is available at <https://www.nature.com/reprints/author-reprints.html>. All co-authors, authors' institutions and

authors' funding agencies can order reprints using the form appropriate to their geographical region.

With best regards,

Jamie

Dr Jamie Horder
Senior Editor
Nature Human Behaviour